# A Canonicalization Perspective on Invariant and Equivariant Learning

**George Ma**[*1]    **Yifei Wang**[*2]    **Derek Lim**[2]    **Stefanie Jegelka**[3]    **Yisen Wang**[4,5†]

[1] School of EECS, Peking University
[2] MIT CSAIL
[3] TUM CIT/MCML/MDSI & MIT EECS/CSAIL
[4] State Key Lab of General Artificial Intelligence,
School of Intelligence Science and Technology, Peking University
[5] Institute for Artificial Intelligence, Peking University

## Abstract

In many applications, we desire neural networks to exhibit invariance or equivariance to certain groups due to symmetries inherent in the data. Recently, frame-averaging methods emerged to be a unified framework for attaining symmetries efficiently by averaging over input-dependent subsets of the group, *i.e.*, frames. What we currently lack is a principled understanding of the design of frames. In this work, we introduce a canonicalization perspective that provides an essential and complete view of the design of frames. Canonicalization is a classic approach for attaining invariance by mapping inputs to their canonical forms. We show that there exists an inherent connection between frames and canonical forms. Leveraging this connection, we can efficiently compare the complexity of frames as well as determine the optimality of certain frames. Guided by this principle, we design novel frames for eigenvectors that are strictly superior to existing methods—some are even optimal—both theoretically and empirically. The reduction to the canonicalization perspective further uncovers equivalences between previous methods. These observations suggest that canonicalization provides a fundamental understanding of existing frame-averaging methods and unifies existing equivariant and invariant learning methods. Code is available at https://github.com/PKU-ML/canonicalization.

## 1 Introduction

When designing machine learning models, incorporating data symmetry provides a strong inductive bias that facilitates learning and generalization [55, 57, 64], as evidenced in multiple applications like convolutional neural networks [30], graph neural networks [52], point clouds [47, 48], *etc*. Often these symmetry priors require models to be invariant or equivariant to certain groups $G$. Among these approaches, model-specific methods restrict every model component to respect data symmetries, which, however, often sacrifices expressive power [68, 39]. On the other hand, model-agnostic methods allow the use of arbitrary (non-invariant) base models, and ensure invariance or equivariance through averaging over group actions [40, 69, 45]. This approach can attain universal expressive power with first-order backbones, but comes with high computation cost for exponentially large groups (*e.g.*, permutation).

---

[*]Equal Contribution. George Ma has graduated from Peking University, and is currently a Ph.D. student at UC Berkeley.
[†]Corresponding Author: Yisen Wang (yisen.wang@pku.edu.cn).

38th Conference on Neural Information Processing Systems (NeurIPS 2024).

To alleviate the latter challenge, frame averaging (FA) [45] has been recognized to be a general framework that can achieve invariance and equivariance efficiently by averaging over a small (input-dependent) subset of the group $\mathcal{F}(X) \subset G$, known as a frame. Frame averaging can improve the averaging complexity of orders of magnitude, and has found wide applications in multiple fields such as graph neural networks [43], materials modeling [13], antibodies generation [38], *etc*. Nevertheless, existing frames are still computationally prohibitive in many domains, in particular, exponentially large for graphs [45] and eigenvectors [36], making it an intriguing problem to explore the design of more efficient frames of lower complexity.

However, a key challenge in this direction is a lack of rigorous ways to characterize the complexity of frames, since existing frames in the literature are still heuristically designed. Although it sounds like a being only of theoretical interest, the ability to characterize the complexity and expressiveness of algorithms has played a vital role in the development of modern algorithms and deep learning models [27]. As an example, the WL hierarchies [65, 54, 72, 71, 73], alongside other complexity measures [46], have been the guideline for developing expressive graph neural networks [17, 39, 74]. However, for averaging methods, we still lack a formal language to quantify, compare, and improve different approaches, which hinders principled development in this area.

In this paper, we propose *canonicalization* as an essential view of frame averaging and a practical yet principled measure for the complexity of frames. Canonicalization is a classical technique with wide applications in graph theory [5], algebra [41] and geometry [4], and it has also recently been explored for learning with symmetry [36]. The key idea of a canonicalization $\mathcal{C}$ is to map all inputs that are equivalent under a group $G$ to the same *canonical form* $\mathcal{C}(X)$. With the canonical input $\mathcal{C}(X)$, any (non-invariant) base neural network $\phi$ will give a $G$-invariant mapping. We show that for a given group, the differences between any frames can be reduced to the differences of their corresponding canonicalizations. Moreover, in contrast to frames that are generally hard to analyze, we show that the canonicalization perspective is much more fertile, leading to a set of theoretical conditions and practical principles for quantifying the complexity and expressiveness of different frames.

To illustrate the benefits of this canonicalization framework, we focus on the symmetries of eigenvectors (sign and basis group), which covers a wide range of applications like graph learning [36, 33], PCA methods [32] and spectral clustering [3]. In this domain, we show that the canonicalization perspective now allows us to answer some long-standing theoretical questions and derive some better or even optimal frames. Specifically, we reveal that for sign invariance (a major concern for eigenvectors), *any* sign-invariant network, including the popular SignNet [33], can be reduced to the *same* input canonicalization, equivalently. Building on this fundamental result, we easily resolve the open problem of SignNet's universality, by showing that it is *not* a universal approximator of functions that are sign invariant and permutation equivariant; we also show how to modify SignNet to be universal with minimal changes. Moreover, as a concrete, practical application, we develop new canonicalizations and frames for eigenvectors, named Orthogonal Axis Projection (OAP), which attain *optimal* or at least *better-than-prior* complexities for unconstrained and constrained scenarios.

At last, we validate our theoretical findings on the EXP dataset, showing that our canonicalization and frames indeed yield orders of lower complexity while demonstrating expressive power beyond 1-WL for distinguishing non-isomorphic graphs. We also show that permutation-equivariant OAP canonicalizes more eigenvectors than previous methods, resulting in better performance on the ZINC and OGBG molecular graph tasks.

## 2   Preliminaries

Let $f \colon \mathcal{X} \to \mathcal{Y}$ be a function and $G$ be a group acting on $\mathcal{X}$ and $\mathcal{Y}$. We say $f$ is *invariant* to $G$ (or $G$-invariant) if for all $g \in G$ and $X \in \mathcal{X}$, we have $f(g \cdot X) = f(X)$. Similarly, we say $f$ is *equivariant* to $G$ (or $G$-equivariant) if for all $g \in G$ and $X \in \mathcal{X}$ we have $f(g \cdot X) = g \cdot f(X)$. We treat invariance as a special case of equivariance by letting $G$ act on $\mathcal{Y}$ as the identity transformation. Existing research has developed neural networks with specific equivariance properties, such as permutation invariance/equivariance [70, 53], rotation invariance/equivariance [62, 60, 63], sign/basis invariance [33], sign equivariance [32], and multi-set equivariance [75].

In the rest of the paper, we define $V, W$ to be vector spaces with norms $\|\cdot\|_V, \|\cdot\|_W$, and $G$ be a group. The elements $g \in G$ act on vectors in $V, W$ with the group's representations $\rho_1 \colon G \to \mathrm{GL}(V)$ and $\rho_2 \colon G \to \mathrm{GL}(W)$, where $\mathrm{GL}(V)$ is the space of invertible linear maps $V \to V$. The group $G$ induces

an equivalence relation on $V$, such that $u \sim v$ if and only if there exists $g \in G$ such that $u = \rho_1(g)v$. For each input $X$, we denote its orbit or equivalence class by $V_G(X) = \{\rho_1(g)^{-1}X \mid g \in G\}$, and its automorphism group as $G_X = \{g \in G \mid \rho_1(g)X = X\}$.

## 2.1 Averaging Methods

Given a network $\phi\colon V \to W$, the naïve way to achieve invariance is through *group averaging*:

$$\Phi_{\mathrm{GA}}(X) = \frac{1}{|G|} \sum_{g \in G} \phi(\rho_1(g)^{-1}X).$$

The resulting $\Phi$ is $G$-invariant, and preserves universal expressive power if $\phi$ is itself universal. Some existing works adopt this approach [40, 69]. However, exact averaging becomes intractable if the cardinality of $G$ is large, for example, the permutation group of graphs. In such cases, random sampling of the group is necessary [40, 14], at the sacrifice of exact symmetry.

Frame averaging [45] is another approach to achieve exact invariance by averaging over a subset of the group on an input $X$, *i.e.,* a frame $\mathcal{F}(X) \subset G$ that maintains $G$-equivariance: $\mathcal{F}(\rho_1(g)X) = g\mathcal{F}(X) = \{gh \mid h \in \mathcal{F}(X)\}$. If an equivariant frame $\mathcal{F}$ is easy to compute, and its cardinality $|\mathcal{F}(X)|$ is not too large, then the following frame averaging scheme

$$\Phi_{\mathrm{FA}}(X; \mathcal{F}, \phi) = \frac{1}{|\mathcal{F}(X)|} \sum_{g \in \mathcal{F}(X)} \phi(\rho_1(g)^{-1}X) \tag{1}$$

also provides the required function symmetrization. $\Phi_{\mathrm{FA}}$ is $G$-invariant, and universally approximates $G$-invariant functions that are approximable by $\phi$. Equivariance can be achieved similarly by multiplying $\rho_2(g)$ with each term. Puny et al. [45] also proposed a variant called *invariant frame averaging* by averaging over the cosets $\mathcal{F}(X)/G_X$ to achieve invariance. Since there is no established way to find the representatives of $\mathcal{F}(X)/G_X$, they adopt uniform sampling from $\mathcal{F}(X)$ to approximate $\Phi_{\mathrm{FA}}$. In Section 3.1, the proposed canonicalization achieves the same complexity without sampling.

Kaba et al. [26] proposed to learn an equivariant canonicalization function for equivariance. They define the canonicalization function to be $h\colon V \to G$ that is $G$-equivariant. Let $\phi$ be the backbone network, then the network taking form $\rho_2(h(X))\phi(\rho_1(h(X))^{-1}X)$ is equivariant and universal. The canonicalization $h$ can be seen as a frame whose output has size 1. However, such a $G$-equivariant $h$ that outputs a single group element does not exist for inputs $X$ with non-trivial automorphism (they introduce a relaxed version of equivariance in this case). Following the classic literature, we define canonicalization on the input space $V$ instead of the group space $G$, which also avoids the problem above.

## 2.2 Symmetries of Eigenvectors

Denote the Laplacian matrix of a graph as $\boldsymbol{L} = \boldsymbol{I} - \hat{\boldsymbol{A}}$, where $\hat{\boldsymbol{A}}$ is the normalized adjacency matrix. Laplacian Positional Encoding (LapPE) uses the eigenvectors of $\boldsymbol{L}$ as positional encoding. LapPE enjoys the benefits of having permutation equivariance and universal expressive power [36], but also suffers from two well-known *ambiguity* problems. The first one, known as *sign ambiguity*, captures that for a unit-norm eigenvector $\boldsymbol{u}_{\lambda_i}$ corresponding to eigenvalue $\lambda_i$, the sign flipped $-\boldsymbol{u}_{\lambda_i}$ is also a unit-norm eigenvector of the same eigenvalue. The second one, termed *basis ambiguity*, captures that eigenvalues with multiplicity degree $d_i > 1$ can have any orthogonal basis in its eigenspace as valid eigenvectors. Because of these ambiguities, we can get distinct GNN outputs for the same graph, resulting in unstable and sub-optimal performance [14, 29, 33]. Besides, sign and basis ambiguities also exist in general eigenvectors that do not require permutation equivariance, which are also widely used in real-world applications, such as PCA methods [32] and spectral clustering [3]. We defer more background to Appendix D.

## 3 Canonicalization: A Unified and Essential View of Equivariant Learning

In this section, we reduce existing averaging methods to canonicalization and show canonicalization can serve as a unified perspective of these methods. Then, using insights from canonicalization theory, we show how SignNet and BasisNet—two invariant networks on eigenvectors—are equivalent to their underlying canonicalizations, while solving the open problem regarding their expressivity.

## 3.1 A Reduction from Frames to Canonicalizations

Frame averaging reduces the number of forward passes in the averaging step compared with group averaging. However, a major obstacle in analyzing the complexity of frames are the automorphisms of the input: $G_X = \{g \in G \mid \rho_1(g)X = X\}$, which can be exponentially large and intractable to compute. For instance, consider defining a frame of a graph as the set of all permutations that sort its node features in increasing order. If there are $m$ nodes in the graph with identical node features, then the frame size is at least $m!$, which grows exponentially. However, all permutations in the frame actually result in the same graph, indicating room for improvement in the efficiency of frame averaging.

In this work, we propose an alternative view to design frames that overcome the difficulties above. For each $X \in V$, instead of averaging over the group elements as in FA (Eq. 1), one can directly average over the *output* elements of the transformations, which is not affected by the complexity of automorphisms since all automorphisms in $G_X$ yield the same output. This converts the problem of finding a $G$-equivariant subset of the group to finding a $G$-invariant set of inputs. In fact, in the classic literature, this problem is known as canonicalization, that achieves invariance by mapping inputs in the same equivalence class to the same canonical form. Formally, we define a *canonicalization* as a set-valued function $\mathcal{C}: V \to 2^V \setminus \varnothing$, such that it is $G$-invariant: $\mathcal{C}(\rho_1(g)X) = \mathcal{C}(X), \forall X \in V, g \in G$. We call its output $\mathcal{C}(X)$ the *canonical form* of $X$. Among possible canonicalizations, we are most interested in *orbit canonicalization*, where the canonical form falls back into the equivalence class: $\mathcal{C}(X) \subset V_G(X)$ for all $X \in V$. With a canonical form, one can instead perform canonical averaging (CA) over $\mathcal{C}(X)$ to obtain invariant[3] representations:

$$\Phi_{\mathrm{CA}}(X; \mathcal{C}, \phi) = \frac{1}{|\mathcal{C}(X)|} \sum\nolimits_{X_0 \in \mathcal{C}(X)} \phi(X_0). \tag{2}$$

The following theorem establishes the equivalence between canonicalizations and frames.

**Theorem 3.1.** *For any frame $\mathcal{F}$ there exists an orbit canonicalization $\mathcal{C}_{\mathcal{F}}$ s.t. for all $X \in V$, $g \in G$, and backbone $\phi$, we have $\Phi_{\mathrm{FA}}(X; \mathcal{F}, \phi) = \Phi_{\mathrm{CA}}(X; \mathcal{C}_{\mathcal{F}}, \phi)$ and*

$$|\mathcal{C}_{\mathcal{F}}(X)| = |\mathcal{F}(X)|/|G_X| \leq |\mathcal{F}(X)|.$$

*In turn, for any orbit canonicalization $\mathcal{C}$ there exists a frame $\mathcal{F}_{\mathcal{C}}$ s.t. for all $X \in V$, $g \in G$, and backbone $\phi$, we have $\Phi_{\mathrm{CA}}(X; \mathcal{C}, \phi) = \Phi_{\mathrm{FA}}(X; \mathcal{F}_{\mathcal{C}}, \phi)$ and*

$$|\mathcal{F}_{\mathcal{C}}(X)| = |G_X| \cdot |\mathcal{C}(X)| \geq |\mathcal{C}(X)|.$$

Theorem 3.1 reveals that frames and canonicalizations have a fundamental equivalence. In other words, a frame is built upon a canonicalization, and a canonicalization induces a frame. Henceforth, we can reduce the difficult problem of designing (equivariant) frames to designing canonicalizations. This simple change of perspective also allows us to have a better theoretical characterization of the complexity and optimality of frames. From now on, we adopt a canonicalization language and reveal some key properties of canonicalization that provide principled guidelines for frame/canonicalization design.

## 3.2 Theoretical Properties of Canonicalization: Universality, Optimality, and Canonicalizability

Here, we examine the theoretical properties of canonicalization in terms of its expressiveness and efficiency. Subsequently, we present unique insights provided by the canonicalization perspective that are not found in the existing literature on frames.

For expressiveness, we define the universality of canonicalization as follows. A canonicalization $\mathcal{C}$ is *universal* if for any $G$-invariant function $f: V \to W$, there exists a well-defined function $\phi: 2^V \to W$ such that $f(X) = \phi(\mathcal{C}(X))$ for all $X \in V$. The following theorem shows that a canonicalization is universal *iff* it corresponds to an orbit canonicalization.

**Theorem 3.2.** *A canonicalization $\mathcal{C}$ is universal iff there exists an orbit canonicalization $\mathcal{C}_c$ and an injective mapping $g: 2^V \to 2^V$ such that $g(\mathcal{C}(X)) = \mathcal{C}_c(X), \forall X \in V$.*

---

[3]We can similarly achieve (relaxed) equivariance [26] by letting the canonicalization output an additional canonizing action $g_0$ with $\rho_1(g_0)X = X_0$, and multiplying $\rho_2(g_0)$ on each term.

For a formal analysis of efficiency, we refer to $|\mathcal{C}(X)|$ as the *complexity* of a canonicalization on $X \in V$. A canonicalization $\mathcal{C}$ is *superior* to another canonicalization $\mathcal{C}'$ if it has smaller complexity on all elements: $|\mathcal{C}(X)| \leq |\mathcal{C}'(X)|, \forall X \in V$. A canonicalization $\mathcal{C}$ is *optimal* if it is superior to any canonicalization. The proposition below establishes the universality and invariance of canonical averaging.

**Theorem 3.3.** *Let $\mathcal{C}$ be an orbit canonicalization. The canonical average $\Phi_{\mathrm{CA}}$ is G-invariant. As long as the backbone network $\phi$ is universal, $\Phi_{\mathrm{CA}}$ is universal in the sense that it can approximate any continuous G-invariant function $f: V \to W$ up to arbitrary precision.*

The canonicalization size $|\mathcal{C}(X)|$ may differ for different inputs. In the most ideal case, an input $X$ admits a single canonical form with $|\mathcal{C}(X)| = 1$, which we call $X$ a *canonicalizable* element. Formally, an element $X \in V$ is *canonicalizable* if there exists an orbit canonicalization $\mathcal{C}$ such that $|\mathcal{C}(X)| = 1$. Otherwise, we call it *uncanonicalizable*, which may happen under additional constraints on the canonicalization. For example, one cannot determine a canonical sign for $[-1, 1]$ when the canonicalization is required to be permutation equivariant, according to Ma et al. [36].

In fact, a major advantage of transforming frames into canonicalization lies in identifying *uncanonicalizable* inputs when additional constraints are imposed, a feature absent in existing literature on frames. In particular, the canonicalizability of inputs offers novel insights into the expressive power of **invariant networks with equivariance constraints**. As will be demonstrated in Section 3.3, invariant networks like SignNet lose expressive power on uncanonicalizable inputs. This enables us to prove the non-universality of SignNet, resolving an open problem in the literature [33, 32]. Canonicalizability is a property of inputs, making it sensible to describe it using canonicalization rather than frames. We refer to Appendix A for a comprehensive discussion on the advantages of the canonicalization perspective.

## 3.3 Reducing Sign-Invariant Networks to Canonicalization

In this section we show how canonicalization can be applied to the eigenvector ambiguity problem, and solve an open question regarding the expressiveness of invariant networks.

There are two general methods to attain exact sign and basis invariance for eigenvectors: SignNet and BasisNet [33] and Laplacian Canonicalization [36]. SignNet is parameterized as the network $f: \mathbb{R}^{n \times k} \to \mathbb{R}^d$ on eigenvectors $\boldsymbol{u}_1, \ldots, \boldsymbol{u}_k$ as

$$f(\boldsymbol{u}_1, \ldots, \boldsymbol{u}_k) = \rho\big([\phi(\boldsymbol{u}_i) + \phi(-\boldsymbol{u}_i)]_{i=1}^k\big),$$

where $\phi$ and $\rho$ are unrestricted neural networks and $[\cdot]_i$ denotes concatenation of vectors. Ma et al. [36] instead achieved sign invariance on canonicalizable input with

$$f(\boldsymbol{u}_1, \ldots, \boldsymbol{u}_k) = \rho\big([\mathrm{MAP}(\boldsymbol{u}_i)]_{i=1}^k\big),$$

where $\mathrm{MAP}$ denotes their canonicalization algorithm MAP, which leverages the sign-invariant and permutation-equivariant projection operator to find a canonical sign. Empirically, the two approaches attain comparable performance in practice. Meanwhile, MAP enjoys better computational efficiency because it only requires input pre-processing while SignNet requires two-branch encoding. Although the two algorithms seem rather different, we prove that any permutation-equivariant and sign-invariant function, including SignNet, is equivalent to a close variant of MAP, which we call $\mathrm{MAP}_{++}$.

**Theorem 3.4.** *A function $h: \mathbb{R}^n \to \mathbb{R}^{n \times d_{\mathrm{out}}}$ is permutation equivariant and sign invariant iff there exists a permutation equivariant (w.r.t. its first input) function $\phi: \mathbb{R}^n \times \{0, 1\} \to \mathbb{R}^{n \times d_{\mathrm{out}}}$ such that*

$$h(\boldsymbol{u}) = \phi\big(\mathrm{MAP}_{++}(\boldsymbol{u}), \mathbb{1}_{\boldsymbol{u}}\big), \forall \boldsymbol{u} \in \mathbb{R}^n,$$

*where* $\mathrm{MAP}_{++}(\boldsymbol{u}) = \begin{cases} \mathrm{MAP}(\boldsymbol{u}), & \text{if } \boldsymbol{u} \text{ is canonicalizable,} \\ |\boldsymbol{u}|, & \text{otherwise,} \end{cases}$ *and $\mathbb{1}_{\boldsymbol{u}} \in \{0, 1\}$ indicates whether $\boldsymbol{u}$ is canonicalizable. Here $|\boldsymbol{u}|$ denotes element-wise absolute value.*

Theorem 3.4 indicates that any sign invariant and permutation equivariant function on a single eigenvector can be reduced to a certain mapping based on the MAP++ canonicalization, which allows a unified characterization for such functions. However, when taking the entire eigenvector matrix as input, processing each eigenvector alone (as in SignNet) will take their absolute values (Theorem 3.4),

which inevitably loses relative position information between different eigenvectors. As a result, both SignNet and MAP++ are not universally expressive. The same result also holds for BasisNet, since SignNet is a special case of first-order BasisNet with multiplicity $d = 1$.[4] The universality of SignNet has been an open problem in the literature [33, 32] and we show that a reduction to the canonicalization perspective can provide a fundamental solution to such problems. Concretely, we also construct two non-isomorphic graphs that SignNet fails to distinguish in Appendix E.5.

**Corollary 3.5.** *SignNet and BasisNet with first-order permutation equivariant $\phi$ [37] cannot universally approximate all permutation-equivariant and sign/basis-invariant functions.*

# 4 Exploring Optimal Canonicalization of Eigenvectors

In this section, we delve into the sign and basis ambiguity problems of eigenvectors and graph positional encodings. We propose novel canonicalization algorithms that are provably superior to existing approaches and even optimal. Specifically, we aim to design a canonicalization algorithm $\mathcal{C}$ operating on eigenvectors $\boldsymbol{U} \in \mathbb{R}^{n \times d}$, that is invariant to sign/basis transformations, (possibly) equivariant to permutation transformations, and outputs a set of eigenvectors $\boldsymbol{U}^* \in \mathbb{R}^{n \times d}$ in the same eigenspace as $\boldsymbol{U}$. We consider two settings: without (Section 4.1) and with (Section 4.2) permutation equivariance, corresponding to different problem scenarios.

## 4.1 Optimal Canonicalization without Permutation

First, we consider the case when we do not need to consider the permutation equivariance of eigenvectors, for example, when samples have a specific ordering. Applications broadly include control systems [50], image segmentation [66], source separation [42], fluid dynamics [25], *etc.*

**Sign Invariance.** Although SignNet can also be applied to such cases, we show that a simple canonicalization that determines directions with the first non-zero entry of the eigenvector $\boldsymbol{u}$ can also achieve sign invariance and permutation equivariance while preserving universality.

---

**Algorithm 1** Canonicalization for eliminating sign ambiguity of eigenvectors

---

**Require:** The eigenvector $\boldsymbol{u} \in \mathbb{R}^n$
**Ensure:** The canonical form $\boldsymbol{u}^*$ of $\boldsymbol{u}$
    Let $i$ be the smallest index such that $u_i \neq 0$
    $\boldsymbol{u}^* \leftarrow \boldsymbol{u}$ if $u_i > 0$, $\boldsymbol{u}^* \leftarrow -\boldsymbol{u}$ otherwise

---

**Basis Invariance.** Inspired by MAP-basis [36], we design a more general and powerful canonicalization based on the Gram-Schmidt Orthogonalization of projection vectors, named *orthogonalized axis projection* (OAP). Specifically, let $\boldsymbol{U} \in \mathbb{R}^{n \times d}$ be eigenvectors in a $d$ dimensional eigenspace, and let $\mathscr{P} = \boldsymbol{U}\boldsymbol{U}^\top$ be the projection matrix onto the eigenspace $\mathrm{span}(\boldsymbol{U})$. Denote $\boldsymbol{e}_1, \ldots, \boldsymbol{e}_n$ as the standard axis vectors of $\mathbb{R}^n$, that is, $\boldsymbol{e}_i$ has 1 at the $i$-th entry and 0 at the other entries. The following Algorithm 2 eliminates basis ambiguities of all eigenvectors.

---

**Algorithm 2** OAP Canonicalization for eliminating basis ambiguity of eigenvectors

---

**Require:** The eigenvectors $\boldsymbol{U} \in \mathbb{R}^{n \times d}$
**Ensure:** The canonical form $\boldsymbol{U}^*$ of $\boldsymbol{U}$
    Let $i_1 < \cdots < i_d$ be the smallest indices *s.t.* $\|\mathscr{P}\boldsymbol{e}_{i_j}\| > 0$ and $\mathscr{P}\boldsymbol{e}_{i_j}$ are linearly independent, $1 \leq j \leq d$
    $\boldsymbol{U}^* \leftarrow \mathrm{GS}(\mathscr{P}\boldsymbol{e}_{i_1}, \ldots, \mathscr{P}\boldsymbol{e}_{i_d})$, where GS denotes Gram-Schmidt Orthogonalization

---

Intuitively, Algorithm 2 finds a set of standard basis vectors with the smallest indices such that their projection on the eigenspace is still a basis. We note that the indices $i_1, \ldots, i_d$ can be found by iteratively checking whether each $\mathscr{P}\boldsymbol{e}_i$ is non-zero and linearly independent with the already-found

---

[4]It is still possible to achieve universality with higher-order networks [33], but they are computationally intractable in practice [37, 49].

projection vectors ($i = 1, \ldots, n$), and adding them if they satisfy these conditions. This can be done in $\mathcal{O}(n^3)$ time (the same as eigendecomposition). The following theorem guarantees that we can always find such indices.

**Theorem 4.1.** *Given a set of eigenvectors $\boldsymbol{U} \in \mathbb{R}^{n \times d}$, let $\mathscr{P} = \boldsymbol{U}\boldsymbol{U}^{\mathrm{T}}$ denote the projection matrix of the eigenspace. Let $\boldsymbol{e}_1, \ldots, \boldsymbol{e}_n$ denote the standard basis vectors. Then, there exists indices $1 \leq i_1 < \cdots < i_d \leq n$, such that for all $1 \leq j \leq d$, we have $\|\mathscr{P}\boldsymbol{e}_{i_j}\| > 0$, and the vectors $\mathscr{P}\boldsymbol{e}_{i_1}, \ldots, \mathscr{P}\boldsymbol{e}_{i_d}$ are linearly independent.*

**Optimality.** We show that Algorithm 1 and 2 are optimal, and can canonicalize *all* eigenvectors.

**Theorem 4.2.** *Algorithm 1 is an optimal orbit canonicalization for all eigenvectors $\boldsymbol{u} \in \mathbb{R}^n$ under sign ambiguity. Algorithm 2 is an optimal orbit canonicalization for all eigenvectors $\boldsymbol{U} \in \mathbb{R}^{n \times d}$ under basis ambiguity.*

These algorithms eliminate ambiguities of eigenvectors, which is useful in many applications. In Appendix D we show their application to achieve orthogonal equivariance in PCA-frame methods.

## 4.2 Better Canonicalization with Permutation

In this section, we further consider the constraint of permutation equivariance when pursuing sign and basis invariance, which is important for certain data structures like graphs.

**Sign Invariance.** We can retain the universality of MAP and SignNet by extending them to perform canonical averaging (Section 3.1) on *uncanonicalizable inputs* with the following MAP-full canonicalization

$$\mathrm{MAP}_{\mathrm{full}}(\boldsymbol{u}) = \begin{cases} \mathrm{MAP}(\boldsymbol{u}), & \text{if } \boldsymbol{u} \text{ is canonicalizable}, \\ \{\boldsymbol{u}, -\boldsymbol{u}\}, & \text{otherwise}. \end{cases} \tag{3}$$

Since Ma et al. [36] proved that MAP canonicalizes all sign-canonicalizable eigenvectors, and if the eigenvector is uncanonicalizable, then the optimal size $|\mathcal{C}(\boldsymbol{u})|$ is 2. Thus, MAP-full is optimal.

**Basis Invariance.** Contrary to the sign case, basis invariance is harder to obtain and MAP-basis proposed by Ma et al. [36] is known to be not optimal. Here, we propose a permutation-equivariant basis canonicalization that is provably superior to MAP-basis. Notice that in OAP (Algorithm 2), the way that we construct smallest indices $i_1, \ldots, i_d$ is not permutation equivariant. This motivates us to extend OAP with a permutation equivariant procedure to determine the indices. Specifically, we adopt the following permutation-equivariant hash function to rank axis projections:

$$\alpha_i = \mathrm{hash}(p_{ii}, \{\!\{p_{ij}\}\!\}_{j \neq i}), \ i = 1, \ldots, n. \tag{4}$$

where $\boldsymbol{p}_i = \mathscr{P}\boldsymbol{e}_i$. According to the number of distinct values in $\{\alpha_i\}$ (denoted as $k$), we divide all standard basis vectors $\{\boldsymbol{e}_i\}$ into $k$ disjoint groups $\mathcal{B}_i$, in descending order of $\alpha_i$. Then we define a summary vector $\boldsymbol{x}_i$ for each group $\mathcal{B}_i$ as $\boldsymbol{x}_i = \sum_{\boldsymbol{e}_j \in \mathcal{B}_i} \boldsymbol{e}_j + c$, where $c$ is a tunable constant. In this way, we arrive at a permutation-equivariant version of OAP for LapPE in Algorithm 3. The summary vectors $\{\boldsymbol{x}_{i_j}\}$ are now permutation equivariant (Appendix B.1), in contrast to $\{\boldsymbol{e}_{i_j}\}$ in Algorithm 2.

---

**Algorithm 3** OAP Canonicalization for eliminating basis ambiguity of Laplacian positional encoding

---

**Require:** The Laplacian eigenvectors $\boldsymbol{U} \in \mathbb{R}^{n \times d}$
**Ensure:** The canonical form $\boldsymbol{U}^*$ of $\boldsymbol{U}$
    Let $i_1 < \cdots < i_d$ be the smallest indices *s.t.* $\|\mathscr{P}\boldsymbol{x}_{i_j}\| > 0$ and $\mathscr{P}\boldsymbol{x}_{i_j}$ are linearly independent, $1 \leq j \leq d$
    $\boldsymbol{U}^* \leftarrow \mathrm{GS}(\mathscr{P}\boldsymbol{x}_{i_1}, \ldots, \mathscr{P}\boldsymbol{x}_{i_d})$, where GS denotes Gram-Schmidt Orthogonalization

---

In contrast to Algorithm 2, the indices $i_1, \ldots, i_d$ may not always exist. The existence and values of these indices can be determined in $\mathcal{O}(n^2 d^2)$ time (*c.f.*, Appendix B.3).

## 4.3 OAP as a Unified Canonicalization for Eigenvectors

Another interesting aspect of OAP is that with a flexible choice of hashing values $\alpha_i$ and indices $i_j$, OAP can serve as a unified framework that encompasses many existing canonicalization algorithms for eigenvectors (as well as graphs).

First, it is easy to see that the sign canonicalizations (Algorithm 1 and MAP-sign) are special cases of their basis versions with $d = 1$. For basis canonicalization, there are two existing methods. One is MAP-basis [36] which also utilizes axis projection but a different construction of basis. Another is proposed in Puny et al. [45] as a way to construct frames for graphs which, according to Theorem 3.1, corresponds to a canonicalization for graphs and can also be adapted to canonicalize LapPE (we defer technical details to Appendix E.8). We call it FA-lap. The following theorem reveals that both MAP and FA-lap are special cases of OAP with degenerated hash functions.

**Theorem 4.3.** *Let $\alpha_i$ $(i = 1, \ldots, n)$ be the outputs of the hash function in the OAP algorithm defined in Equation 4, and let $i_j$ $(j = 1, \ldots, d)$ be the indices found in Algorithm 3. Then, the MAP algorithm [36] is equivalent to the OAP algorithm that takes $\alpha_i = \|\mathscr{P}_i\|$ for all $1 \leq i \leq n$ and $i_j = j$ for all $1 \leq j \leq d$. The FA-lap algorithm [45] is equivalent to the OAP algorithm that takes $\alpha_i = \mathscr{P}_{ii}$ for all $1 \leq i \leq n$.*

From this connection, we can see that the hashes of MAP and FA-lap are not as distinctive as OAP with Eq. 4. As a result, OAP can canonicalize more eigenvectors and is strictly superior to MAP and FA-lap. There is no superiority between MAP and FA-lap. It remains an open problem whether the permutation-equivariant OAP (Algorithm 3) is optimal.

Similarly, these canonicalizations can be adapted to canonicalize graphs (which we call MAP-graph, FA-graph, OAP-graph), and have the same complexity hierarchy (more details in Appendix E.8). We summarize them as follows.

**Corollary 4.4.** *For eigenvectors, OAP is strictly superior to MAP and FA-lap. For graphs, OAP-graph is strictly superior to MAP-graph and FA-graph.*

## 5 Experiments

In this section, we evaluate the expressive power and efficiency of CA on the EXP dataset; we apply our canonicalization to the $n$-body problem for orthogonal equivariance; we also evaluate OAP for LapPE on graph regression tasks.

### 5.1 Expressive Power and Frame Size

To validate the universal expressiveness of CA in Theorem 3.3, we conduct experiments on EXP [1], which is designed to explicitly evaluate the expressiveness of GNNs. It consists of pairs of graphs that are non-isomorphic but 1-WL indistinguishable. We follow the setup of Balcilar et al. [7]. As baselines, we use GCN [67], GAT [61], GIN [68] and ChebNet [59], **all of which are permutation equivariant**. We also equip GIN with unique node IDs and apply FA/CA on them, which we denote as FA-GIN+ID and CA-GIN+ID. Note that node IDs endow GIN with universality [40] while also breaking permutation equivariance, and FA/CA restore permutation equivariance while preserving universality. Results are shown in Table 1. All MP-GNNs achieve trivial performance since their expressive power is limited by 1-WL [68]. Both FA and CA achieve perfect accuracy, showing expressiveness beyond 1-WL.

Table 1: Accuracy on EXP.

| Model | Accuracy |
| --- | --- |
| GCN | 50% |
| GAT | 50% |
| GIN | 50% |
| ChebNet | 82% |
| FA-GIN+ID | 100% |
| CA-GIN+ID | 100% |

To evaluate the efficiency of different methods, we also compare their average frame size $|\mathcal{F}(X)|$ or canonicalization size $|\mathcal{C}(X)|$ on EXP. As shown in Table 2, OAP-graph is more efficient than FA-graph, verifying Corollary 4.4. Since $|\mathcal{C}(X)| < |\mathcal{F}(X)|$, CA is more efficient than FA, validating Theorem 3.1. We note that although the numbers are still very large in Table 2 and we still need sampling, a smaller frame/canonicalization size would lead to faster convergence rate towards the true average. This is proven in Appendix C.

### 5.2 Graph Regression and Classification

We evaluate OAP on ZINC [24]. We measure the ratio of non-canonicalizable[5] eigenvectors among all eigenvectors by FA-lap, MAP, and OAP. As shown in Table 3, they are equivalent in addressing sign

---

[5]Here "non-canonicalizable" means that the canonicalization algorithm fails to canonicalize the eigenvectors.

Table 2: The average frame size (F) and canonicalization size (C) on EXP with two canonicalization algorithms: FA-graph and OAP-graph.

| Method | Avg $|\mathcal{F}(X)|$ | Avg $|\mathcal{C}(X)|$ | F/C Ratio |
|---|---|---|---|
| FA-graph | $2.10 \times 10^{24}$ | $5.84 \times 10^{21}$ | **360×** |
| OAP-graph | $2.55 \times 10^{22}$ | $7.57 \times 10^{19}$ | **337×** |
| FA/OAP Ratio | **82×** | **77×** | |

ambiguity, while OAP has the least ratio of non-canonicalizable eigenvectors under basis ambiguity, aligning with Corollary 4.4.

We conduct experiments on ZINC [24] and OGBG [22] (*c.f.*, Appendix H). We use GatedGCN [9] and PNA [12] as backbones and apply different PE methods: (1) No positional encoding; (2) Laplacian PE combined with random sign (RS) that randomly flips the signs of eigenvectors [14]; (3) SignNet [33]; (4) MAP [36]; (5) OAP; (6) OAP with LSPE layers [15]. On ZINC, we also evaluate the FA-GIN+ID model in Section 5.1, as well as GIN with edge features (*i.e.*, GINE) [21]. Methods implemented by ourselves are marked with *. The results are reported in Table 4, 5 and 6.

Table 3: Ratio of non-canonicalizable eigenvectors on ZINC.

| Canonicalization | FA-lap | MAP | OAP |
|---|---|---|---|
| Sign | 5.4% | 5.4% | 5.4% |
| Basis | 2.2% | 1.0% | **0.2%** |

Table 4: Results on ZINC with 500K parameter budget. All scores are averaged over 4 runs with 4 different seeds.

| Model | PE | $k$ | #Param | MSE $\downarrow$ |
|---|---|---|---|---|
| GIN | ID + FA | 0 | 495K | 0.613 ± 0.023 |
| GINE | ID + FA | 0 | 495K | 0.546 ± 0.048 |
| GatedGCN | None | 0 | 504K | 0.251 ± 0.009 |
| | LapPE + RS | 8 | 505K | 0.202 ± 0.006 |
| | SignNet | 8 | 495K | 0.121 ± 0.005 |
| | MAP | 8 | 486K | 0.120 ± 0.005 |
| | OAP | 8 | 473K | **0.118 ± 0.0006** |
| | OAP + LSPE | 8 | 491K | **0.098 ± 0.0009** |
| PNA | None | 0 | 369K | 0.141 ± 0.004 |
| | LapPE + RS | 8 | 474K | 0.132 ± 0.010 |
| | SignNet | 8 | 476K | 0.105 ± 0.007 |
| | MAP | 8 | 462K | 0.101 ± 0.005 |
| | OAP | 8 | 462K | **0.098 ± 0.0008** |
| | OAP + LSPE | 8 | 549K | **0.095 ± 0.004** |

Table 5: Results on MOLTOX21. All scores are averaged over 4 runs with 4 different seeds.

| Model | PE | $k$ | #param | ROCAUC $\uparrow$ |
|---|---|---|---|---|
| GatedGCN | None | 0 | 1004K | 0.772 ± 0.006 |
| | LapPE + RS | 3 | 1004K | 0.774 ± 0.007 |
| | SignNet* | 3 | 1367K | 0.773 ± 0.003 |
| | MAP | 3 | 1505K | 0.784 ± 0.005 |
| | OAP | 3 | 1542K | **0.787 ± 0.005** |
| PNA | None | 0 | 5245K | 0.755 ± 0.008 |
| | LapPE + RS | 16 | 2453K | 0.756 ± 0.009 |
| | SignNet* | 16 | 1754K | 0.750 ± 0.009 |
| | MAP | 16 | 1951K | 0.761 ± 0.002 |
| | OAP | 16 | 1950K | **0.768 ± 0.002** |

Table 6: Results on MOLPCBA. All scores are averaged over 4 runs with 4 different seeds.

| Model | PE | $k$ | #param | AP $\uparrow$ |
|---|---|---|---|---|
| GatedGCN | None | 0 | 1008K | 0.262 ± 0.001 |
| | LapPE + RS | 3 | 1009K | 0.266 ± 0.002 |
| | SignNet* | 3 | 2415K | 0.260 ± 0.002 |
| | MAP | 3 | 2658K | 0.268 ± 0.002 |
| | OAP | 3 | 2658K | **0.270 ± 0.002** |
| PNA | None | 0 | 6551K | 0.279 ± 0.003 |
| | LapPE + RS* | 16 | 6423K | 0.275 ± 0.004 |
| | SignNet* | 16 | 4493K | OOM |
| | MAP | 16 | 4612K | **0.281 ± 0.003** |
| | OAP | 16 | 4612K | 0.279 ± 0.002 |

In Table 4, we observe that FA achieves the lowest performance since it breaks permutation equivariance. The results presented in Tables 4, 5 and 6 demonstrate that incorporating LapPE leads to improved performance across nearly all cases compared to no PE, highlighting the advantages of leveraging expressive PEs. Notably, SignNet, MAP, and OAP show significant performance gains over LapPE, emphasizing the benefits of addressing ambiguities of Laplacian eigenvectors. OAP performs best among these methods, consistent with our theoretical expectations of its superiority over MAP and SignNet. Additionally, SignNet experiences memory issues even with fewer parameters in Table 6, underscoring the increased memory demands associated with incorporating invariant network architectures. Furthermore, in Table 4, we find that LSPE enhances the performance of OAP across all models.

We compare the time and memory of canonicalization methods with their non-FA backbone on ZINC in Table 7. Using canonicalization algorithms only increases the pre-processing time of the backbone, which is negligible compared to the training time. On the other hand, the two-branch architecture of SignNet increases the training time and memory.

Table 7: Comparison of time and memory of canonicalization methods with their non-FA backbone on ZINC. For the backbone models, the node features are first concatenated with positional encodings and fed to a positional encoding network (we use masked GIN in our experiments), then the outputs of the positional encoding network are used as input for the main network (GatedGCN or PNA). For the SignNet models, the positional encoding network is substituted with SignNet, which has a two-branch architecture. For models with MAP and OAP, the positional encodings are canonicalized before fed to the positional encoding network.

| Model | Pre-processing time | Training time | Total time | Memory |
|---|---|---|---|---|
| GatedGCN backbone | - | 3h26min | 3h26min | 1860MiB |
| GatedGCN + SignNet | 30.03s | 4h13min | 4h13min | 2124MiB |
| GatedGCN + MAP | 133.67s | 3h20min | 3h22min | 1850MiB |
| GatedGCN + OAP | 186.38s | 3h25min | 3h28min | 1860MiB |
| PNA backbone | - | 16h31min | 16h31min | 2242MiB |
| PNA + SignNet | 30.03s | 18h1min | 18h1min | 2570MiB |
| PNA + MAP | 133.67s | 16h47min | 16h49min | 2244MiB |
| PNA + OAP | 186.38s | 14h54min | 14h57min | 2312Mib |

## 6 Conclusion and Discussion

In this paper, we illustrated canonicalization as a useful view of frames. From this perspective, we established concrete theoretical conditions for determining the complexity of frames, as well as deriving better and even optimal frames. We believe that the canonicalization perspective has the potential to unify different invariant and equivariant learning approaches for a unified characterization.

One limitation of this work lies in that we do not fully resolve the optimality of eigenvector canonicalization under permutation equivariance. It is still yet unknown whether the proposed OAP canonicalization is optimal for Laplacian eigenvectors, and there is still much to explore in terms of canonicalization algorithms across other domains.

## Acknowledgement

Yisen Wang was supported by National Key R&D Program of China (2022ZD0160300), National Natural Science Foundation of China (92370129, 62376010), and Beijing Nova Program (20230484344, 20240484642). Derek Lim was supported by an NSF Graduate Fellowship. Yifei Wang and Stefanie Jegelka were supported by NSF AI Institute TILOS (NSF CCF-2112665), NSF award 2134108, and the Alexander von Humboldt Foundation.

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

# Appendix

## Table of Contents

# A    Advantages of the canonicalization perspective

In this paper, we introduced canonicalization as a new perspective to learning with symmetry. We demonstrated how shifting from frames to canonicalization could offer valuable new perspectives, enabling us to address the open problem of the universality of SignNet and BasisNet. Theorem 3.1 establishes an equivalence between frames and canonicalization, prompting the question: why should we favor canonicalization over frames? We put forward three primary reasons:

1. Frames encounter challenges with symmetric inputs, *i.e.* those with non-trivial automorphism, like graphs. In such cases, the automorphism group of an input could grow exponentially, consequently expanding the frame size exponentially as well. However, in theory, each graph has a single canonicalization, highlighting the potential for improvement in frames when dealing with invariance. Here we provide a concrete example concerning node features of graphs. As illustrated in Figure 1, each node in the graph possesses a unique node ID and a color representing its feature. Suppose we define the frame of this graph as the set of all permutations sorting the node features in the order of grey, blue, and green. For instance, one permutation might yield the sorted node IDs: 0, 1, 4, 6, 8, 3, 5, 2, 7. Although the frame size for this graph is calculated as $5! \times 2! \times 2! = 480$, applying elements of the frame results in the same graph. Consequently, the frame size exceeds the canonicalization size by a factor of $480$.

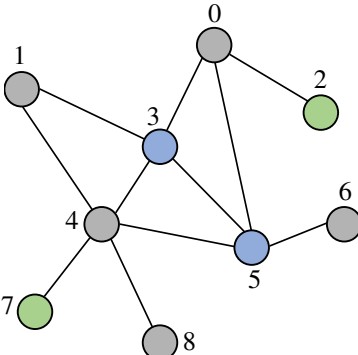

Figure 1: An example where the frame size exceeds the canonicalization by a factor of $480$.

2. Frames must adhere to $G$-equivariance: $\mathcal{F}(\rho_1(g)X) = g\mathcal{F}(X)$, posing a stringent requirement. Consequently, constructing frames becomes challenging. While one approach involves constructing a canonicalization first and then inducing its corresponding frame, a preferable option is to directly utilize canonicalization. The presence of input automorphism complicates the analysis and improvement of existing frames. An optimal canonicalization maintains a size of $1$ for all inputs, irrespective of input symmetry. In contrast, the size of even optimal frames depends on input automorphism.

3. The principal advantage of canonicalization over frame lies in the presence of *uncanonicalizable* inputs, offering intriguing theoretical insights. Sign invariance serves as a notable example. According to Ma et al. [36], an eigenvector $\boldsymbol{u} \in \mathbb{R}^n$ is uncanonicalizable with respect to sign *iff* there exists a permutation matrix $\boldsymbol{P}$ such that $\boldsymbol{u} = -\boldsymbol{P}\boldsymbol{u}$. In Theorem 3.4, we further demonstrated that invariant networks applied to Laplacian eigenvectors lose expressive power on such inputs. Since canonicalizability pertains to the input space $V$ rather than $G$, analyzing the expressiveness of invariant networks through their underlying canonicalization is considerably more convenient.

    We aim to delve deeper into this insight by exploring the concept of canonicalizability as introduced in Ma et al. [36]. As demonstrated therein, uncanonicalizable inputs emerge when equivariance constraints are imposed on an already invariant canonicalization while ensuring its universality is maintained. It is the trilemma of invariance, equivariance, and universality that gives rise to uncanonicalizable inputs. A frame possesses equivariance inherently, and examining the frame alone suggests that imposing additional equivariance constraints would not result in conflicts, as observed in the input space.

# B   Implementation and verification of MAP

## B.1   The Gram-Schmidt process

The Gram-Schmidt process is heavily used in our canonicalization algorithms; we discuss it in this subsection. The Gram-Schmidt process is a method of constructing an orthonormal basis from a set of vectors in an inner product space, most commonly the Euclidean space $\mathbb{R}^n$ equipped with the standard inner product. Specifically, the Gram-Schmidt process takes a finite, linearly independent set of vectors $\{x_1, x_2, \ldots, x_m\}$ in an inner product space $V$ with $m \leq n$, and generates an orthogonal set of vectors $\{v_1, v_2, \ldots, v_m$ that spans the same $m$-dimensional subspace of $\mathbb{R}^n$, where

$$v_1 = x_1,$$

$$v_2 = x_2 - \frac{\langle x_2, v_1 \rangle}{\|v_1\|^2} v_1,$$

$$v_3 = x_3 - \frac{\langle x_3, v_1 \rangle}{\|v_1\|^2} v_1 - \frac{\langle x_3, v_2 \rangle}{\|v_2\|^2} v_2,$$

$$\vdots$$

$$v_i = x_i - \sum_{k=1}^{i-1} \frac{\langle x_i, v_k \rangle}{\|v_k\|^2} v_k,$$

$$\vdots$$

$$v_m = x_m - \sum_{k=1}^{m-1} \frac{\langle x_m, v_k \rangle}{\|v_k\|^2} v_k.$$

The Gram-Schmidt process can be implemented in PyTorch [44] using QR decomposition.

```python
def orthogonalize(U: Tensor) -> Tensor:
    Q, R = torch.linalg.qr(U)
    S = torch.sign(torch.diag(R))
    return Q * S
```

If the input matrix $U \in \mathbb{R}^{n \times d}$ has full rank (in our case it does because $U$ is a matrix of eigenvectors), and if we require the diagonal elements of $R$ to be positive, then $Q$ and $R$ are unique, and (column vectors of) $Q \in \mathbb{R}^{n \times d}$ gives the Gram-Schmidt orthogonalization of (column vectors of) $U$. The matrix $S$ in the above code is to ensure that diagonal elements of $RS$ are positive.

Proof for the correctness of this method can be found in common linear algebra textbooks [11]. Here we are mainly interested in the equivariance properties of this method. In particular, we prove that it is permutation equivariant and orthogonally equivariant.

**Theorem B.1.** *The Gram-Schmidt process is permutation equivariant and orthogonally equivariant.*

*Proof.* We first prove the permutation equivariance of GS by induction. Let $P \in \mathbb{R}^{n \times n}$ be an arbitrary permutation matrix. Let $\{x_1', x_2', \ldots, x_m'\}$ be defined by $x_i' = P x_i$ for $1 \leq i \leq m$, and let $\{v_1', v_2', \ldots, v_m'\}$ be the orthogonal basis generated from $\{x_1', x_2', \ldots, x_m'\}$ by GS. Clearly $v_1' = x_1' = P x_1$ is permutation equivariant. Suppose $v_1', \ldots, v_{k-1}'$ are all permutation equivariant, then

$$v_k' = x_k' - \sum_{j=1}^{k-1} \frac{\langle x_k', v_j' \rangle}{\|v_j'\|^2} v_j' = P x_k - \sum_{j=1}^{k-1} \frac{\langle P x_k, P v_j \rangle}{\|v_j\|^2} P v_j = P v_k,$$

where the last equality holds because the inner product operator is invariant under permutation. Thus $v_k$ is also permutation equivariant, and by induction we conclude that GS is permutation equivariant.

The orthogonal equivariance of GS can be proved similarly. Let $Q \in \mathbb{R}^{n \times n}$ be an arbitrary orthogonal matrix. Let $\{x_1', x_2', \ldots, x_m'\}$ be defined by $x_i' = Q x_i$ for $1 \leq i \leq m$, and let $\{v_1', v_2', \ldots, v_m'\}$ be the orthogonal basis generated from $\{x_1', x_2', \ldots, x_m'\}$. Clearly $v_1'$ is orthogonally equivariant.

Suppose $\boldsymbol{v}'_1, \ldots, \boldsymbol{v}'_{k-1}$ are all orthogonally equivariant, then

$$\boldsymbol{v}'_k = \boldsymbol{x}'_k - \sum_{j=1}^{k-1} \frac{\langle \boldsymbol{x}'_k, \boldsymbol{v}'_j \rangle}{\|\boldsymbol{v}'_j\|^2} \boldsymbol{v}'_j = \boldsymbol{Q}\boldsymbol{x}_k - \sum_{j=1}^{k-1} \frac{\langle \boldsymbol{Q}\boldsymbol{x}_k, \boldsymbol{Q}\boldsymbol{v}_j \rangle}{\|\boldsymbol{v}_j\|^2} \boldsymbol{Q}\boldsymbol{v}_j = \boldsymbol{Q}\boldsymbol{v}_k$$

is also orthogonally equivariant, where the last equality holds because orthogonal transformations preserve the inner product. Thus GS is also orthogonally equivariant. $\square$

Recall that in Algorithm 3, we first defined a set of values $\alpha_i = \mathrm{hash}(p_{ii}, \{\!\{ p_{ij} \}\!\}_{j \neq i})$ for $i = 1, \ldots, n$, where $\boldsymbol{p}_i = \mathscr{P}\boldsymbol{x}_i$ is the projection of standard basis vectors onto the eigenspace $V$. Suppose there are $k$ distinct values in $\{\alpha_i\}$, i.e., $|\{\alpha_i\}| = k$. Then according to the values of $\alpha_i$ we divided the standard basis vectors $\{\boldsymbol{e}_i\}$ into $k$ disjoint groups $\mathcal{B}_i$ (arranged in descending order of the values of $\alpha_i$), and defined a summary vector $\boldsymbol{x}_i$ for each group $\mathcal{B}_i$ as the sum of standard basis vectors in that group. Consider the projections of these $k$ summary vectors, $\mathscr{P}\boldsymbol{x}_i$. Let $i_1 < i_2 < \cdots < i_d$ be the smallest $d$ indices such that $\|\mathscr{P}\boldsymbol{x}_{i_j}\| > 0$ and $\mathscr{P}\boldsymbol{x}_{i_j}$ are linearly independent, $1 \leq j \leq d$. In the final step, we applied the Gram-Schmidt process to $\mathscr{P}\boldsymbol{x}_{i_1}, \ldots, \mathscr{P}\boldsymbol{x}_{i_d}$ to obtain the resulting canonical form. These steps can be summarized as follows.

1. Compute $\alpha_i = \mathrm{hash}(p_{ii}, \{\!\{ p_{ij} \}\!\}_{j \neq i})$, $1 \leq i \leq n$.

2. Define summary vectors $\boldsymbol{x}_i = \sum \boldsymbol{e}_j$, where the summands $\boldsymbol{e}_j$ share the same $\alpha_j$ value, and the summary vectors are arranged in descending order of these values.

3. Find the smallest indices $i_1 < \cdots < i_d$ such that $\|\mathscr{P}\boldsymbol{x}_{i_j}\| > 0$ and $\mathscr{P}\boldsymbol{x}_{i_j}$ are linearly independent, $1 \leq j \leq d$.

4. Apply GS to $\{\mathscr{P}\boldsymbol{x}_{i_1}, \ldots, \mathscr{P}\boldsymbol{x}_{i_d}\}$ to obtain the final canonical form.

In this algorithm we mainly make use of the permutation equivariance property of GS to construct an overall permutation equivariant canonicalization, while the basis invariance of this canonicalization relies on the basis invariance of projections themselves. We prove that Algorithm 3 is permutation equivariant.

**Theorem B.2.** *The OAP canonicalization in Algorithm 3 is permutation equivariant.*

*Proof.* Consider the canonical forms of $\boldsymbol{U}$ and $\boldsymbol{U}' = \boldsymbol{P}\boldsymbol{U}$, where $\boldsymbol{P}$ is an arbitrary permutation matrix. We analyze each step of this algorithm. The permutation $\boldsymbol{P}$ acts on the eigenspace by $\mathscr{P}' = \boldsymbol{P}\mathscr{P}\boldsymbol{P}^\top$, as well as on the standard basis vectors by $\boldsymbol{e}'_i = \boldsymbol{P}\boldsymbol{e}_i$.

1. In step 1, since each projection $\boldsymbol{p}'_i = \mathscr{P}'\boldsymbol{e}'_i = \boldsymbol{P}\mathscr{P}\boldsymbol{P}^\top \boldsymbol{P}\boldsymbol{e}_i = \boldsymbol{P}\boldsymbol{p}_i$ is permutation equivariant, the value vector $\boldsymbol{\alpha} = (\alpha_1, \ldots, \alpha_n)$ is also permutation equivariant.

2. In step 2, we summed vectors $\boldsymbol{e}_i$ with the same corresponding $\alpha_i$ to get the summary vectors, while the summary vectors themselves are arranged in descending order of the corresponding $\alpha_i$. If we arrange the entries of $\boldsymbol{\alpha}$ in descending order as

$$\alpha_{i_1} \geq \alpha_{i_2} \geq \cdots \geq \alpha_{i_n},$$

then the indices vector $\boldsymbol{i} = (i_1, \ldots, i_n)$ would be "almost" permutation equivariant, except that indices corresponding to equal entries could have any ordering. However, as we summed up all $\boldsymbol{e}_i$ with equal corresponding $\alpha_i$, the summary vectors $\boldsymbol{x}_i$ would be exactly permutation equivariant for $1 \leq i \leq k$, in the sense that $\boldsymbol{x}'_i = \boldsymbol{P}(\sum_{\boldsymbol{e}_j \in \mathcal{B}_i} \boldsymbol{e}_j) = \boldsymbol{P}\boldsymbol{x}_i$.

3. The projections $\mathscr{P}\boldsymbol{x}_{i_j}$ are themselves permutation equivariant. Since the norm operator $\|\cdot\|$ and the linear independence of vectors are not affected by permutation, the indices $i_1 < \cdots < i_d$ are permutation invariant.

4. Using the permutation equivariance of the Gram-Schmidt process (Theorem B.1), we conclude that the whole algorithm is permutation equivariant.

$\square$

## B.2 Complete pseudo-code of Algorithm 2

In Algorithm 2, we proposed the OAP-eig canonicalization to address the basis ambiguity of eigenvectors without permutation equivariance. The complete pseudo-code of OAP-eig, along with a detailed time complexity analysis, is shown in Algorithm 4.

---

**Algorithm 4** OAP-eig canonicalization for eliminating basis ambiguity without permutation equivariance

---

**Require:** The eigenvectors $U \in \mathbb{R}^{n \times d}$
**Ensure:** The canonical form $U^*$ of $U$
    $\mathscr{P} \leftarrow UU^\top$                                           ▷ $\mathcal{O}(n^2 d)$ complexity
    $U_{\text{span}} \leftarrow$ empty matrix of shape $n \times 0$
    $r \leftarrow 0$                                                      ▷ rank of $U_{\text{span}}$
    **for** [ **do**$\mathcal{O}(n^2 d^2)$ complexity]$i = 1, \ldots, n$
        $u \leftarrow \text{normalize}(\mathscr{P}_i)$                              ▷ $\mathcal{O}(n)$ complexity
        **if** $\|u\| < \varepsilon$ **then**                ▷ floating-point errors are considered
            **continue**
        **end if**
        $U_{\text{tmp}} \leftarrow [U_{\text{span}}, u^\top]$
        **if** $\text{rank}(U_{\text{tmp}}) = r + 1$ **then**     ▷ $u$ is linearly independent with $U_{\text{span}}$, $\mathcal{O}(nd^2)$ complexity
            $U_{\text{span}} \leftarrow U_{\text{tmp}}$
            $r \leftarrow r + 1$
            **if** $r = d$ **then**                         ▷ found indices $i_1, \ldots, i_d$
                **break**
            **end if**
        **end if**
    **end for**
    $U^* \leftarrow \text{GS}(U_{\text{span}})$                                      ▷ $\mathcal{O}(nd^2)$ complexity
    **return** $U^*$

---

The overall time complexity of Algorithm 4 is $\mathcal{O}(n^2 d^2)$. Since in real-world datasets the value of $d$ is usually small (Figure 2 in Ma et al. [36]), Algorithm 4 is more efficient than eigendecomposition itself whose time complexity is $\mathcal{O}(n^3)$.

In the for loop of Algorithm 4, we aim to find indices $i_1, \ldots, i_d$ such that $\|\mathscr{P}e_{i_j}\| = \|\mathscr{P}_{i_j}\| > 0$ and $\mathscr{P}_{i_j}$ are linearly independent for $1 \leq j \leq d$. We look for these indices in a iterative fashion. The matrix $U_{\text{span}}$ contains all the already-found indices. We initialize $U_{\text{span}}$ to be an empty matrix of shape $n \times 0$, and every time we find an index $i$ such that $\|\mathscr{P}_i\| > 0$ and $\mathscr{P}_i$ is linearly independent with the already-found columns vectors of $U_{\text{span}}$, we concatenate the normalized $\mathscr{P}_i$ to $U_{\text{span}}$. Once we have found a total of $d$ indices (denoted as $r$ in Algorithm 4), we break out of the loop and return the Gram-Schmidt orthogonalization of $U_{\text{span}}$. By the properties of GS, $U^*$ is still a set of orthonormal basis in the eigenspace. However, as we iterate through $\mathscr{P}e_1, \mathscr{P}e_2, \ldots, \mathscr{P}e_n$, the algorithm is not permutation equivariant (the standard basis vectors $\{e_i\}$ are permuted by permutations).

## B.3 Complete pseudo-code of Algorithm 3

In Algorithm 3, we proposed the OAP-lap canonicalization to address the basis ambiguity of Laplacian eigenvectors while respecting permutation equivariance. The complete pseudo-code of OAP-lap, along with a detailed time complexity analysis, is shown in Algorithm 5.

The overall time complexity of Algorithm 5 is $\mathcal{O}(n^2 d^2)$, which is also more efficient than eigendecomposition itself. Although Algorithm 5 need to be applied to every eigenspace of the graph, we note that the number of multiple eigenvalues (eigenvalues with multiplicity greater than 1 and thus have basis ambiguity) are usually small in real-world datasets (Table 9 in Ma et al. [36]).

Algorithm 5 consists of two parts. In the first part, we try to divide the standard basis vectors $\{e_i\}_{1 \leq i \leq n}$ into disjoint groups $\{\mathcal{B}_i\}_{1 \leq i \leq k}$. To do that, we first calculate a set of values $\alpha_i$, $1 \leq i \leq n$ for each standard basis vector. Any set of permutation-equivariant $\alpha_i$ would guarantee the correctness of the algorithm, but to ensure the algorithm canonicalizes as many eigenvectors as possible (*i.e.*, the *superiority* of canonicalization), we would like the values of $\alpha_i$ to have enough "distinguishability"

**Algorithm 5** OAP-lap canonicalization for eliminating basis ambiguity with permutation equivariance

---

**Require:** The Laplacian eigenvectors $U \in \mathbb{R}^{n \times d}$
**Ensure:** The canonical form $U^*$ of $U$

$\quad \mathscr{P} \leftarrow UU^\top$ $\qquad\qquad\qquad\qquad\qquad\qquad\qquad\qquad$ ▷ $\mathcal{O}(n^2 d)$ complexity

$\quad \boldsymbol{\alpha} \leftarrow \left(\text{hash}(\mathscr{P}_{11}, \{\!\{\mathscr{P}_{1j}\}\!\}_{j \neq 1}), \ldots, \text{hash}(\mathscr{P}_{nn}, \{\!\{\mathscr{P}_{nj}\}\!\}_{j \neq n})\right)$ $\qquad$ ▷ $\mathcal{O}(n^2)$ complexity

$\quad val, ind \leftarrow \text{sort}(\boldsymbol{\alpha})$ $\qquad\qquad\qquad\qquad\qquad\qquad\qquad\qquad$ ▷ $\mathcal{O}(n \log n)$ complexity

$\quad val \leftarrow \text{unique}(val)$ $\qquad\qquad\qquad\qquad\qquad$ ▷ unique values in $\{\alpha_i\}$, $\mathcal{O}(n)$ complexity

$\quad k \leftarrow |val|$

$\quad$ **if** $k < d$ **then**

$\quad\quad$ **break** $\qquad\qquad\qquad\qquad\qquad\qquad\qquad\qquad$ ▷ impossible to find $d$ indices

$\quad$ **end if**

$\quad$ **for** $i = 1, \ldots, k$ **do** $\qquad\qquad\qquad\qquad\qquad\qquad\qquad\qquad$ ▷ $\mathcal{O}(n)$ complexity

$\quad\quad \boldsymbol{x}_i \leftarrow \sum_j \boldsymbol{e}_{ind[j]}$ such that $\alpha_{ind[j]} = val[i]$ $\qquad$ ▷ sum of standard basis vectors in group $\mathcal{B}_i$

$\quad\quad \boldsymbol{x}_i \leftarrow \boldsymbol{x}_i + c\mathbf{1}$ $\qquad\qquad\qquad\qquad\qquad$ ▷ the summary vector of group $\mathcal{B}_i$

$\quad$ **end for**

$\quad U_{\text{span}} \leftarrow$ empty matrix of shape $n \times 0$

$\quad r \leftarrow 0$ $\qquad\qquad\qquad\qquad\qquad\qquad\qquad\qquad$ ▷ rank of $U_{\text{span}}$

$\quad$ **for** $i = 1, \ldots, k$ **do** $\qquad\qquad\qquad\qquad\qquad\qquad\qquad$ ▷ $\mathcal{O}(n^2 d^2)$ complexity

$\quad\quad \boldsymbol{u} \leftarrow \text{normalize}(\mathscr{P}\boldsymbol{x}_i)$ $\qquad\qquad\qquad\qquad\qquad\qquad$ ▷ $\mathcal{O}(n)$ complexity

$\quad\quad$ **if** $\|\boldsymbol{u}\| < \epsilon$ **then** $\qquad\qquad\qquad\qquad\qquad$ ▷ floating-point errors are considered

$\quad\quad\quad$ **continue**

$\quad\quad$ **end if**

$\quad\quad U_{\text{tmp}} \leftarrow [U_{\text{span}}, \boldsymbol{u}^\top]$

$\quad\quad$ **if** $\text{rank}(U_{\text{tmp}}) = r + 1$ **then** $\qquad$ ▷ $\boldsymbol{u}$ is linearly independent with $U_{\text{span}}$, $\mathcal{O}(nd^2)$ complexity

$\quad\quad\quad U_{\text{span}} \leftarrow U_{\text{tmp}}$

$\quad\quad\quad r \leftarrow r + 1$

$\quad\quad\quad$ **if** $r = d$ **then** $\qquad\qquad\qquad\qquad\qquad\qquad$ ▷ found indices $i_1, \ldots, i_d$

$\quad\quad\quad\quad$ **break**

$\quad\quad\quad$ **end if**

$\quad\quad$ **end if**

$\quad$ **end for**

$\quad U^* \leftarrow \text{GS}(U_{\text{span}})$ $\qquad\qquad\qquad\qquad\qquad\qquad\qquad$ ▷ $\mathcal{O}(nd^2)$ complexity

$\quad$ **return** $U^*$

---

(*i.e.*, the values of $\alpha_i$ divide the standard basis vectors into more fine-grained groups, in Babai et al. [6] this is called *refinement*). In Algorithm 5 we define $\alpha_i = \text{hash}(\mathscr{P}_{ii}, \{\!\{\mathscr{P}_{ij}\}\!\}_{j \neq i})$. We proved the permutation equivariance of such $\alpha_i$ in Appendix B.1, and in Appendix E we will prove that they are more fine-grained than previous methods, such as $\alpha_i = \mathscr{P}_{ii}$ in Puny et al. [45] and $\alpha_i = \|\mathscr{P}_i\|$ in Ma et al. [36].

Once we have the values of $\alpha_i$, we can divide $\{\boldsymbol{e}_i\}$ into disjoint groups $\mathcal{B}_i$ accordingly. Each value $\alpha_i$ corresponds to a standard basis vector $\boldsymbol{e}_i$. Two standard basis vectors belong to the same group if and only if their corresponding $\alpha_i$ are equal, and different groups are arranged in descending order of the values of $\alpha_i$. For each group, we define its summary vector $\boldsymbol{x}_i$ to be the sum of standard basis vectors in that group, for $1 \leq i \leq k$. In the second part of Algorithm 5, we use the same method as Algorithm 4 to find the indices $i_1, \ldots, i_d$, except that the vectors $\{\mathscr{P}\boldsymbol{e}_i\}_{1 \leq i \leq n}$ are substituted with $\{\mathscr{P}\boldsymbol{x}_i\}_{1 \leq i \leq k}$. Note that in contrast to Algorithm 4, now these vectors are permutation-equivariant by construction.

## B.4 Verifying the correctness of Algorithm 2

The correctness of Algorithm 1 is obvious and we skip that part. In this subsection we verify the correctness of Algorithm 2 through random simulation. Here "correctness" refers to two things: for any eigenvectors $U \in \mathbb{R}^{n \times d}$ the indices $i_1, \ldots, i_d$ can always be found; and that the whole algorithm is basis-invariant as desired. The verification program is shown in Algorithm 6. Let $U \in \mathbb{R}^{n \times d}$ be a random orthonormal matrix representing the eigenvectors, and let $Q \in \text{O}(d)$ be a random orthonormal matrix that rotates the eigenvectors in their corresponding eigenspace. We pass $U$ and

$UQ$ to Algorithm 2 (denoted as canonicalize) and compare the results. If our algorithm is correct, $U$ and $UQ$ should produce identical outputs.

---

**Algorithm 6** Verify the correctness of Algorithm 2

---

$correct \leftarrow 0$, $total \leftarrow 0$
**for** $i = 1, 2, \ldots, trials$ **do**
    $n \leftarrow$ a random positive integer (greater than 1)
    $d \leftarrow$ a random positive integer (less than $n$)
    $U \leftarrow$ a random orthonormal matrix in $\mathbb{R}^{n \times d}$
    $U_0 \leftarrow$ canonicalize($U$)
    $Q \leftarrow$ a random orthonormal matrix in $\mathbb{R}^{d \times d}$
    $W \leftarrow UQ$
    $W_0 \leftarrow$ canonicalize($W$)
    $correct \leftarrow correct + 1$ **if** $|U_0 - W_0| < \varepsilon$           ▷ test basis invariance
    $total \leftarrow total + 1$
**end for**
**print** the values of $correct$ and $total$

---

We conduct 1000 trials. The results are $correct = total = 1000$, showing the basis invariance of Algorithm 2. The function canonicalize will throw an error and terminate if the indices $i_1, \ldots, i_d$ (and thus the canonical form) cannot be found, so the successful execution of this program also shows that we can always find such indices on random eigenvectors. This aligns with the conclusion of Theorem 4.1.

We can also modify Algorithm 6 to verify the orthogonal equivariance of canonical averaging in the $n$-body problem. Let model denote the model with canonical averaging. The program is shown in Algorithm 7. The results show that canonical averaging achieves exact orthogonal equivariance.

---

**Algorithm 7** Verify the orthogonal equivariance of canonical averaging in the $n$-body problem

---

$correct \leftarrow 0$, $total \leftarrow 0$
**for** $i = 1, 2, \ldots, trials$ **do**
    $X \leftarrow$ node features in the dataset, representing the inital location of particles
    $X_0 \leftarrow$ model($X$)
    $Q \leftarrow$ a random orthonormal matrix in $\mathbb{R}^{d \times d}$, where $d$ is the dimension of node features
    $W \leftarrow XQ$
    $W_0 \leftarrow$ model($W$)
    $correct \leftarrow correct + 1$ **if** $|X_0 Q - W_0| < \varepsilon$         ▷ test orthogonal equivariance
    $total \leftarrow total + 1$
**end for**
**print** the values of $correct$ and $total$

---

### B.5 Verifying the correctness of Algorithm 3

In this subsection we verify the correctness of Algorithm 3. Here "correctness" refers to the permutation equivariance and basis invariance of the canonicalization algorithm. The verification program is shown in Algorithm 8. Let $U \in \mathbb{R}^{n \times d}$ be a random orthonormal matrix representing the Laplacian eigenvectors, $P \in \mathbb{R}^{n \times n}$ be a random permutation matrix that permutes the rows of $U$, and let $Q \in O(d)$ be a random orthonormal matrix that rotates the Laplacian eigenvectors in their corresponding eigenspace. We pass $U, PU, UQ, PUQ$ to Algorithm 3 (denoted as canonicalize) and compare the results. If our algorithm is correct, $UQ$ should produce invariant outputs, while $PU$ and $PUQ$ should produce equivariant outputs.

We conduct 1000 trials. The results are $p\_correct = q\_correct = pq\_correct = total = 1000$, showing the permutation equivariance and basis invariance of Algorithm 3. The function canonicalize will throw an error and terminate if it could not find a canonical form for $U$. While for LapPE uncanonicalizable eigenvectors do exist [36], the successful execution of this program show that empirically the probability for random Laplacian eigenvectors to be uncanonicalizable is 0. Ma et al. [36] has a more rigorous discussion on this in Appendix H of their paper.

**Algorithm 8** Verify the correctness of Algorithm 3

$p\_correct \leftarrow 0$, $q\_correct \leftarrow 0$, $pq\_correct \leftarrow 0$, $total \leftarrow 0$
**for** $i = 1, 2, \ldots, trials$ **do**
    $n \leftarrow$ a random positive integer (greater than 1)
    $d \leftarrow$ a random positive integer (less than $n$)
    $\boldsymbol{U} \leftarrow$ a random orthonormal matrix in $\mathbb{R}^{n \times d}$
    $\boldsymbol{U}_0 \leftarrow \text{canonicalize}(\boldsymbol{U})$
    $\boldsymbol{P} \leftarrow$ a random permutation matrix
    $\boldsymbol{V} \leftarrow \boldsymbol{PU}$
    $\boldsymbol{V}_0 \leftarrow \text{canonicalize}(\boldsymbol{V})$
    $p\_correct \leftarrow p\_correct + 1$ **if** $|\boldsymbol{PU}_0 - \boldsymbol{V}_0| < \varepsilon$           ▷ test permutation equivariance
    $\boldsymbol{Q} \leftarrow$ a random orthonormal matrix in $\mathbb{R}^{d \times d}$
    $\boldsymbol{W} \leftarrow \boldsymbol{UQ}$
    $\boldsymbol{W}_0 \leftarrow \text{canonicalize}(\boldsymbol{W})$
    $q\_correct \leftarrow q\_correct + 1$ **if** $|\boldsymbol{U}_0 - \boldsymbol{W}_0| < \varepsilon$           ▷ test basis invariance
    $\boldsymbol{Y} \leftarrow \boldsymbol{PW}$
    $\boldsymbol{Y}_0 \leftarrow \text{canonicalize}(\boldsymbol{Y})$
    $pq\_correct \leftarrow pq\_correct + 1$ **if** $|\boldsymbol{PU}_0 - \boldsymbol{Y}_0| < \varepsilon$           ▷ test both
    $total \leftarrow total + 1$
**end for**
**print** the values of $p\_correct$, $q\_correct$, $pq\_correct$ and $total$

## C Concentration inequality for drawing without replacement

In the EXP experiment the frame or canonicalization size is very large (Table 2), so we need to randomly sample from the frame or canonicalization. We use the average of model outputs on the sampled graphs to approximate the true average value on all graphs. In this section we prove that a smaller frame/canonicalization size will lead to faster convergence rate towards the true average. Thus, according to Table 2, CA is more sample efficient than FA, and OAP is more sample efficient than FA-graph.

Denote by $\mathcal{F}$ a frame averaging scheme, and by $\mathcal{C}$ the canonicalization induced by $\mathcal{F}$. Let $V_{\mathcal{F}}(X) = \{\!\{\rho_1(g)^{-1} X \mid g \in \mathcal{F}(X)\}\!\}$ be all graphs generated by the frame $\mathcal{F}(X)$ from the input $X$ (in our EXP experiment, $X$ would be a graph in the dataset). Note that $V_{\mathcal{F}}(X)$ is a multi-set, so if $k$ group actions in $\mathcal{F}(X)$ result in the same graph when applied to $X$, then this graph is counted $k$ times in $V_{\mathcal{F}}(X)$. Let $\mathcal{C}(X)$ be the canonicalization of input $X$. Let $\phi$ be the backbone neural network. Sampling from frame/canonicalization works as follows:

1. If the frame size $\mathcal{F}(X)$ is large, then the exact averaging $\frac{1}{|\mathcal{F}(X)|} \sum_{g \in \mathcal{F}(X)} \phi\big(\rho_1(g)^{-1} X\big)$ is intractable. In this case, we randomly sample $n$ graphs $X_1, \ldots, X_n$ from $V_{\mathcal{F}}(X)$ and use the empirical average $\frac{1}{n} \sum_{i=1}^{n} \phi(X_i)$ as an approximation to the true average.

2. Similarly, if the canonicalization size $\mathcal{C}(X)$ is large, then the exact averaging $\frac{1}{|\mathcal{C}(X)|} \sum_{X_0 \in \mathcal{C}(X)} \phi(X_0)$ is intractable. In this case, we randomly sample $n$ graphs $X_1, \ldots, X_n$ from $\mathcal{C}(X)$ and use the empirical average $\frac{1}{n} \sum_{i=1}^{n} \sum_{i=1}^{n} \phi(X_i)$ as an approximation to the true average.

When we sample from $V_{\mathcal{F}}(X)$ or $\mathcal{C}(X)$, there are two approaches: drawing with replacement and drawing without replacement. However, it is well-known that drawing without replacement is preferable, as illustrated in the following theorem.

**Theorem C.1** (Hoeffding [19]). *Let $\mathcal{X} = (x_1, \ldots, x_N)$ be a finite population of $N$ real points, $X_1, \ldots, X_n$ denote a random sample without replacement from $\mathcal{X}$ and $Y_1, \ldots, Y_n$ denote a random sample with replacement from $\mathcal{X}$. If $f : \mathbb{R} \to \mathbb{R}$ is continuous and convex, then*

$$\mathbb{E}f\left(\sum_{i=1}^{n} X_i\right) \leq \mathbb{E}f\left(\sum_{i=1}^{n} Y_i\right).$$

In Theorem C.1, we can understand $f$ as the error between the empirical average and the true average, and $\mathcal{X}$ as the outputs of $\phi$ on the frame or canonicalization. Then we should adopt drawing without replacement since it leads to smaller approximation error and thus faster concentration rate.

When drawing without replacement, we compare the concentration rate between frame averaging $\mathcal{F}$ and the corresponding canonicalization $\mathcal{C}$. By Theorem 3.1 we have $|V_{\mathcal{F}}(X)| = |\mathcal{C}(X)| \cdot |G_X|$, where $G_X$ is the stabilizer of $X$ in group $G$. In fact $V_{\mathcal{F}}(X)$ is just $\mathcal{C}(X)$ repeated for $|G_X|$ times. To see this, we can divide $\mathcal{F}(X)$ into equivalence classes $\mathcal{F}(X)/G_X$, and denote by $\sim$ the corresponding equivalence relation. For two group actions $g_1, g_2 \in \mathcal{F}(X)$, $g_1 \sim g_2$ if and only if there exists $g_0 \in G_X$ such that $g_1 = g_0 g_2$. Since $X = \rho_1(g_0)X$ by definition, group actions in the same equivalence group will result in the same graph when applied to $X$. Since each equivalence group has size $|G_X|$, the multi-set of graphs generated by the frame $V_{\mathcal{F}}(X)$ is just the repetition of $\mathcal{C}(X)$ for $|G_X|$ times.

Intuitively, the frame size $|V_{\mathcal{F}}(X)|$ is $|G_X|$ times larger than the canonicalization size $|\mathcal{C}(X)|$, thus drawing without replacement on $V_{\mathcal{F}}(X)$ would concentrate slower than on $\mathcal{C}(X)$. This is shown in the following theorem.

**Theorem C.2** (Bardenet and Maillard [8]). *Let $\mathcal{X} = (x_1, \ldots, x_N)$ be a finite population of $N > 1$ real points, and $(X_1, \ldots, X_n)$ be a list of size $n < N$ sampled without replacement from $\mathcal{X}$. Then for all $\varepsilon > 0$, the following concentration bounds hold:*

$$\Pr\left[\max_{n \leq k \leq N-1} \frac{\sum_{t=1}^{k}(X_t - \mu)}{k} \geq \varepsilon\right] \leq \exp\left(-\frac{2n\varepsilon^2}{(1 - n/N)(1 + 1/n)(b-a)^2}\right),$$

*where $a = \min_{1 \leq i \leq N} x_i$, $b = \max_{1 \leq i \leq N} x_i$, and $\mu = \frac{1}{N}\sum_{i=1}^{N} x_i$.*

In Theorem C.2, let $\mu$ be the true average and let $X_1, \ldots, X_n$ be the outputs of $\phi$ on the sampled graphs from the frame or canonicalization. The upper bound in Theorem C.2 decreases as the frame/canonicalization size $N$ decreases, indicating faster concentration rate. Since $|\mathcal{C}(X)| < |V_{\mathcal{F}}(X)|$, we conclude that CA is more sample efficient than FA; and since the frame size of OAP-graph is smaller than that of FA-graph, we conclude that OAP-graph is more sample efficient than FA-graph.

# D PCA-frame methods for orthogonal equivariance

## D.1 PCA-frame methods

We can achieve orthogonal equivariance by using PCA-frame methods [32]. Given input $\boldsymbol{X} \in \mathbb{R}^{n \times k}$, we compute orthonormal eigenvectors $\boldsymbol{R_X} \in \mathrm{O}(k)$ of the covariance matrix $\mathrm{cov}(\boldsymbol{X}) = (\boldsymbol{X} - \frac{1}{n}\mathbf{1}\mathbf{1}^\top\boldsymbol{X})^\top(\boldsymbol{X} - \frac{1}{n}\mathbf{1}\mathbf{1}^\top\boldsymbol{X})$, where $\mathrm{O}(k)$ is the group of orthogonal matrices in $\mathbb{R}^{k \times k}$. Then we transform $\boldsymbol{X}$ into its canonical form $\boldsymbol{X}\boldsymbol{R_X}$ and feed it into a network $h$. Finally, we apply $\boldsymbol{R_X}^\top$ to the output of the network to transform it back to its original orientation. The whole process can be described as a neural network model

$$f(\boldsymbol{X}) = h(\boldsymbol{X}\boldsymbol{R_X})\boldsymbol{R_X}^\top.$$

See Figure 2 for an illustration. Intuitively, this first transforms $\boldsymbol{X}$ by $\boldsymbol{R_X}$ into a nearly canonical orientation that is unique up to sign flips; this can be seen as writing the points in the principal components basis, or aligning the principal components of $\boldsymbol{X}$ with the coordinate axes. Then we process $\boldsymbol{X}\boldsymbol{R_X}$ using the model $h$, and finally incorporate orientation information back into the output by post-multiplying by $\boldsymbol{R_X}^\top$.

However, due to the ambiguities of eigenvectors, the canonical form $\boldsymbol{X}\boldsymbol{R_X}$ is not unique, and $f$ cannot achieve exact orthogonal equivariance without addressing these ambiguities. Puny et al. [45] averages over all possible sign choices for $\boldsymbol{R_X}$, which requires $2^k$ forward passes through the base model $h$. Lim et al. [32] resolves these ambiguities with special sign equivariant architectures for $h$. In Section 4.1, we propose to address them using optimal canonicalization.

## D.2 Experiments on the $n$-body problem

We use canonicalization to achieve exact orthogonal equivariance with PCA-frame methods, as introduced in Appendix D. We consider simulating $n$-body problems Fuchs et al. [18]. For baselines

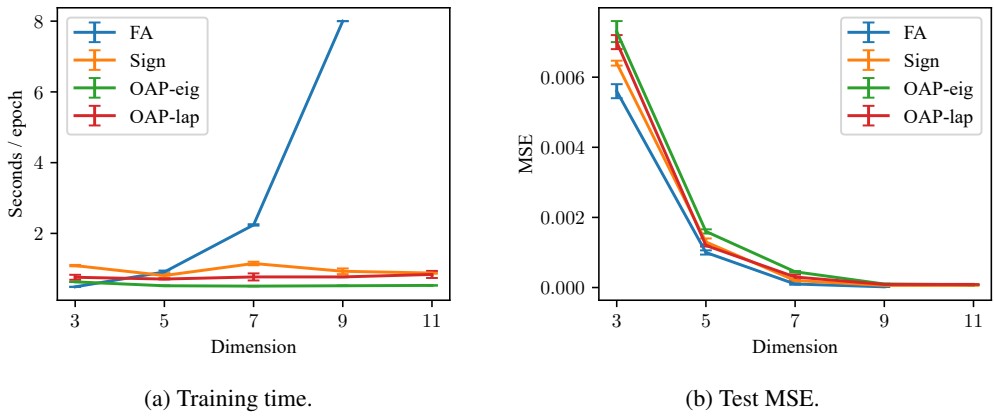

Figure 2: Using PCA-frame methods to achieve orthogonal equivariance, described as a neural network model $f(X) = h(XR_X)R_X^\top$, where $R_X$ is a choice of principal components for the point cloud $X \in \mathbb{R}^{n \times k}$. We first transform $X$ via $R_X$ into an orientation that is unique up to sign flips, then process $XR_X$ using a network $h$, and finally reintegrate orientation information back into the output via $R_X^\top$. Figure reproduced with permission from Lim et al. [32].

we consider FA over signs of eigenvectors and sign-equivariant models [32]. We also evaluate CA based on Algorithm 2 (OAP-eig) and Algorithm 3 (OAP-lap) on this task. We scale the dimension $d$ of the problem and report the training time and MSE of different models. In Figure 3a, we observe that the training time of FA scales exponentially, since it needs to average $2^d$ sign choices. FA runs out of memory on a 32GB V100 GPU for $d = 11$. Both sign-equivariant model and CA have almost constant training time as $d$ increases, while CA is slightly more efficient since it enforces no extra equivariance restraint on the model. In Figure 3b, we observe comparable performance between these methods. CA achieves much better scalability than FA with a small loss in accuracy.



(a) Training time.

(b) Test MSE.



Figure 3: The training time and test MSE of models in the $n$-body problem. Results are averaged over 4 runs with different seeds.

# E   Proofs

## E.1   Proof of Theorem 3.1

*Proof.* Let $\mathcal{F}$ be a frame. Consider the following set-valued function $\mathcal{C}_{\mathcal{F}} \colon V \to 2^V \backslash \varnothing$ defined by

$$\mathcal{C}_{\mathcal{F}}(X) = \{\rho_1(g)^{-1}X \mid g \in \mathcal{F}(X)\}.$$

Then $\mathcal{C}_{\mathcal{F}}$ is $G$-invariant because for any $g' \in G$,

$$\begin{aligned}
\mathcal{C}_{\mathcal{F}}\big(\rho_1(g')X\big) &= \{\rho_1(g)^{-1}\rho_1(g')X \mid g \in \mathcal{F}\big(\rho_1(g')X\big)\} \\
&= \{\rho_1(g)^{-1}\rho_1(g')X \mid g \in g'\mathcal{F}(X)\} \\
&= \{\rho_1(g)^{-1}\rho_1(g')^{-1}\rho_1(g')X \mid g \in \mathcal{F}(X)\} \\
&= \{\rho_1(g)^{-1}X \mid g \in \mathcal{F}(X)\} \\
&= \mathcal{C}_{\mathcal{F}}(X).
\end{aligned}$$

Thus $\mathcal{C}_\mathcal{F}$ is a canonicalization. Clearly it is also an orbit canonicalization since $\mathcal{C}_\mathcal{F}(X) \subset V_G(X)$ for all $X$.

Let $g_1, g_2 \in \mathcal{F}(X)$. We claim that $\rho_1(g_1)^{-1}X = \rho_1(g_2)^{-1}X$ if and only if there exists $g_0 \in G_X$ such that $g_1 = g_0 g_2$, where $G_X = \{g \in G \mid \rho_1(g)X = X\}$ is the stabilizer of $X$. On the one hand, if $g_1 = g_0 g_2$, then $\rho_1(g_1)^{-1}X = \rho_1(g_2)^{-1}\rho_1(g_0)^{-1}X = \rho_1(g_2)^{-1}X$. On the other hand, if $\rho_1(g_1)^{-1}X = \rho_1(g_2)^{-1}X$, then $X = \rho_1(g_1)\rho_1(g_2)^{-1}X$, which means $g_1 g_2^{-1} \in G_X$. Denote $g_1 g_2^{-1} = g_0$, then $g_1 = g_0 g_2$.

Thus, we can divide the frame into equivalence classes $\mathcal{F}(X)/G_X$. Group actions in the same equivalence class result in the same element when applied to $X$, while group actions in different equivalence classes result in different elements. The total number of elements obtained by applying $\mathcal{F}(X)$ to $X$ is $|\mathcal{F}(X)|/|G_X|$, i.e., $|\mathcal{C}_\mathcal{F}(X)| = |\mathcal{F}(X)|/|G_X| \leq |\mathcal{F}(X)|$. If $X$ has non-trivial automorphism, then $|\mathcal{C}_\mathcal{F}(X)| < |\mathcal{F}(X)|$, meaning canonicalization is strictly more efficient than the frame.

Similarly, let $\mathcal{C}$ be an orbit canonicalization. Consider the following set-valued function $\mathcal{F}_\mathcal{C}: V \to 2^G \backslash \varnothing$ defined by

$$\mathcal{F}_\mathcal{C}(X) = \{g \in G \mid \rho_1(g)^{-1}X \in \mathcal{C}(X)\}.$$

Since $\mathcal{C}(X) \subset V_G(X)$, $\mathcal{F}_\mathcal{C}(X)$ is well-defined. It is also $G$-equivariant, since for any $g' \in G$,

$$
\begin{aligned}
\mathcal{F}_\mathcal{C}\big(\rho_1(g')X\big) &= \big\{g \in G \mid \rho_1(g)^{-1}\rho_1(g')X \in \mathcal{C}\big(\rho_1(g')X\big)\big\} \\
&= \{g \in G \mid \rho_1(g'^{-1}g)^{-1}X \in \mathcal{C}(X)\} \\
&= \{g \in g'G \mid \rho_1(g)^{-1}X \in \mathcal{C}(X)\} \\
&= g'\mathcal{F}_\mathcal{C}(X).
\end{aligned}
$$

Thus $\mathcal{F}_\mathcal{C}$ is a frame. We can also see that the canonicalization induced by $\mathcal{F}_\mathcal{C}$ is just $\mathcal{C}$ itself. Repeating the arguments in the first part of this proof, we conclude that $|\mathcal{F}_\mathcal{C}(X)| = |G_X| \cdot |\mathcal{C}(X)| \geq |\mathcal{C}(X)|$. If $X$ has non-trivial automorphism, then canonicalization is strictly more efficient than the frame. $\quad\square$

**Corollary E.1.** *A frame $\mathcal{F}$ is optimal iff the canonicalization that it induces is optimal.*

*Proof.* Corollary E.1 immediately follows from the definition of optimality and the relation $|\mathcal{F}(X)| = |G_X| \cdot |\mathcal{C}(X)|$, where $\mathcal{F}$ and $\mathcal{C}$ induce each other. $\quad\square$

## E.2 Proof of Theorem 3.2

We first show that an orbit canonicalization itself is universal.

**Lemma E.2.** *An orbit canonicalization $\mathcal{C}: V \to 2^V \backslash \varnothing$ is universal.*

*Proof.* Recall universality means that for any $G$-invariant function $f: V \to W$, there exists a well-defined function $\phi: 2^V \to W$ such that $f(X) = \phi(\mathcal{C}(X))$ for all $X \in V$. Since $f$ is $G$-invariant, all inputs in the same equivalence class induced by $G$ produce the same output when fed to $f$. This means that for fixed $X$, $f(X')$ is constant for all $X' \in V_G(X)$. We can simply take the $\phi$ function to be $\phi = f$. Since $\mathcal{C}(X) \subset V_G(X)$, we have $\phi(\mathcal{C}(X)) = f(\mathcal{C}(X)) = f(X)$ for all $X$, which proves the universality of $\mathcal{C}$. $\quad\square$

Then we give the proof of Theorem 3.2.

*Proof. Sufficiency.* If $g(\mathcal{C}(X)) = \mathcal{C}_c(X)$ for some orbit canonicalization $\mathcal{C}_c$ and all $X \in V$, we prove $\mathcal{C}$ is universal. By Lemma E.2, for any $G$-invariant function $f: V \to W$, there exists a well-defined function $\phi_c$ such that $f(X) = \phi_c(\mathcal{C}_c(X))$ for all $X \in V$. Letting $\phi = \phi_c \circ g$, we have $f(X) = \phi_c(g(\mathcal{C}(X))) = \phi(\mathcal{C}(X))$ for all $X \in V$, hence $\mathcal{C}$ is universal.

*Necessity.* If $\mathcal{C}$ is universal, we construct $g: 2^V \to 2^V$ such that it maps $\mathcal{C}(X)$ to any subset of $V_G(X)$, for all $X \in V$. Since $\mathcal{C}$ is invariant, this mapping is well-defined; and since the equivalence classes $\{V_G(X)\}$ are disjoint, this mapping is also injective. Clearly, the canonicalization defined by $\mathcal{C}_c(X) := g(\mathcal{C}(X))$ is an orbit canonicalization. $\quad\square$

### E.3 Proof of Theorem 3.3

*Proof.* The invariance of $\Phi_{\mathrm{CA}}$ is derived from the invariance of $\mathcal{C}$ (in fact this does not rely on $\mathcal{C}$ to be an orbit canonicalization). For all $g \in G$ and $X \in V$,

$$\Phi_{\mathrm{CA}}\big(\rho_1(g)X; \mathcal{C}, \phi\big) = \frac{1}{|\mathcal{C}(\rho_1(g)X)|} \sum_{X_0 \in \mathcal{C}(\rho_1(g)X)} \phi(X_0)$$

$$= \frac{1}{|\mathcal{C}(X)|} \sum_{X_0 \in \mathcal{C}(X)} \phi(X_0) = \Phi_{\mathrm{CA}}(X; \mathcal{C}, \phi).$$

For universality, the $G$-invariance of $f$ implies that the value of $f$ is constant on each equivalence class $V_G(X)$. Since $\mathcal{C}$ is an orbit canonicalization, this further implies that $f$ is constant on each canonicalization $\mathcal{C}(X)$. Thus $f(X) = \Phi_{\mathrm{CA}}(X; \mathcal{C}, f)$ for all $X$. For an arbitrary $G$-invariant function $f$, the approximation error of $\Phi_{\mathrm{CA}}$ is bounded by

$$\|f(X) - \Phi_{\mathrm{CA}}(X; \mathcal{C}, \phi)\|_W = \|\Phi_{\mathrm{CA}}(X; \mathcal{C}, f) - \Phi_{\mathrm{CA}}(X; \mathcal{C}, \phi)\|_W$$

$$= \left\| \frac{1}{|\mathcal{C}(X)|} \sum_{X_0 \in \mathcal{C}(X)} f(X_0) - \frac{1}{|\mathcal{C}(X)|} \sum_{X_0 \in \mathcal{C}(X)} \phi(X_0) \right\|_W$$

$$\leq \frac{1}{|\mathcal{C}(X)|} \sum_{X_0 \in \mathcal{C}(X)} \|f(X_0) - \phi(X_0)\|_W$$

$$\leq \max_{X \in V} \|f(X) - \phi(X)\|_W.$$

Thus the approximation error of $\Phi_{\mathrm{CA}}$ is bounded by that of $\phi$, and the universality of $\phi$ implies the universality of $\Phi_{\mathrm{CA}}$. $\qquad\square$

A direct corollary is that we can universally approximate any $G$-invariant function using MLPs under mild conditions.

**Corollary E.3.** *Let $V, W$ be compact normed linear spaces. For any continuous $G$-invariant $f: V \to W$ and any $\varepsilon > 0$, there exists an MLP network $\phi: V \to W$ such that $\|f(X) - \Phi_{\mathrm{CA}}(X; \mathcal{C}, \phi)\| \leq \varepsilon$ holds for all $X \in V$.*

*Proof.* Since MLP is universal in approximating continuous functions $f: V \to W$ [20], it also universally approximates such $f$ that is $G$-invariant. Corollary E.3 immediately follows from Theorem 3.3. $\qquad\square$

### E.4 Proof of Theorem 3.4

Notice that in Theorem 3.4, the function $\phi$ have two inputs. Apart from the canonical form $\mathrm{MAP}_{++}(\boldsymbol{u})$, it also takes an indicator $\mathbb{1}_{\boldsymbol{u}}$ as input, which indicates whether $\boldsymbol{u}$ is canonicalizable. Thus our proof will be divided into two parts, for canonicalizable and uncanonicalizable eigenvectors respectively.

**Lemma E.4.** *A function $h: \mathbb{R}^n \to \mathbb{R}^{n \times d_{\mathrm{out}}}$ is permutation equivariant and sign invariant on all canonicalizable inputs $\boldsymbol{u} \in \mathbb{R}^n$ iff there exists a permutation equivariant function $\phi: \mathbb{R}^n \to \mathbb{R}^{n \times d_{\mathrm{out}}}$ such that $h(\boldsymbol{u}) = \phi(\mathrm{MAP}_{++}(\boldsymbol{u}))$.*

*Proof. Sufficiency.* If $\phi$ is permutation equivariant, since $\mathrm{MAP}_{++}$ is also permutation equivariant, so is $\phi(\mathrm{MAP}_{++}(\boldsymbol{u}))$. Since $\mathrm{MAP}_{++}$ is a canonicalization for sign, $\phi(\mathrm{MAP}_{++}(\boldsymbol{u}))$ is also sign-invariant.

*Necessity.* Given a permutation equivariant and sign invariant function $h$, we simple take $\phi = h$, where $\phi$ is defined on all canonical forms output by $\mathrm{MAP}_{++}$. Since $\mathrm{MAP}_{++}$ is universal on canonical inputs (because it is an orbit canonicalization), by Theorem 3.3, such $\phi$ exists and $\phi(\mathrm{MAP}_{++}(\boldsymbol{u}))$ is sign invariant, while $\phi$ does not have to be sign invariant. Since $h$ is permutation equivariant, so is $\phi$; and since $\mathrm{MAP}_{++}$ is itself permutation equivariant, so is the entire function $\phi(\mathrm{MAP}_{++}(\boldsymbol{u}))$. $\qquad\square$

Before moving on to the second part, we first introduce some lemmas about the structure of uncanonicalizable eigenvectors under sign ambiguity.

**Lemma E.5.** *If $\boldsymbol{u}$ is uncanonicalizable, then all non-zero values in $\boldsymbol{u}$ are paired opposite values, i.e., the multi-set of all non-zero values in $\boldsymbol{u}$ takes the form $\{\!\!\{a_1, -a_1, a_2, -a_2, \ldots, a_k, -a_k\}\!\!\}$ for some $k$.*

*Proof.* Because $\boldsymbol{P}\boldsymbol{u} = -\boldsymbol{u}$, all non-zero values in $\boldsymbol{u}$ have to be permuted by $\boldsymbol{P}$. For any orbit of the permutation $n_1 \to n_2 \to \cdots \to n_k \to n_1$, from the fact $\boldsymbol{P}\boldsymbol{u} = -\boldsymbol{u}$, we know

$$u_{n_2} = -u_{n_1}, \ u_{n_3} = -u_{n_2} = u_1, \ \ldots, \ u_{n_k} = (-1)^{k+1} u_1 = -u_{n_1}.$$

Thus, $k$ has to be even, and all values in the same orbit of the permutation are paired opposite values of the same magnitude, *e.g.*, $\{\!\!\{u_1, -u_1\}\!\!\}$. Applying it to all orbits of the permutation finishes the proof. $\qquad\square$

**Lemma E.6.** *For any uncanonicalizable eigenvector $\boldsymbol{u}$, there exists a pairwise permutation matrix $\boldsymbol{P}$ (which only sweeps two non-zero opposite numbers) such that $\boldsymbol{P}\boldsymbol{u} = -\boldsymbol{u}$.*

*Proof.* Since the non-zero values in $\boldsymbol{u}$ appear in pairs, we simply choose $\boldsymbol{P}$ such that it sweeps these pairs and keeps all 0's unmoved. $\qquad\square$

Now for the second part.

**Lemma E.7.** *A function $h \colon \mathbb{R}^n \to \mathbb{R}^{n \times d_{\mathrm{out}}}$ is permutation equivariant and sign invariant on all uncanonicalizable inputs $\boldsymbol{u} \in \mathbb{R}^n$ iff there exists a permutation equivariant function $\phi \colon \mathbb{R}^n \to \mathbb{R}^{n \times d_{\mathrm{out}}}$ such that $h(\boldsymbol{u}) = \phi(|\boldsymbol{u}|)$.*

*Proof. Sufficiency.* First, it is obvious that for any $\boldsymbol{u}$, $h(\boldsymbol{u}) = \phi(|\boldsymbol{u}|)$ is sign invariant. It is also permutation equivariant because for any permutation matrix $\boldsymbol{P}$, $h(\boldsymbol{P}\boldsymbol{u}) = \phi(|\boldsymbol{P}\boldsymbol{u}|) = \phi(\boldsymbol{P}|\boldsymbol{u}|) = \boldsymbol{P}\phi(|\boldsymbol{u}|) = \boldsymbol{P}h(\boldsymbol{u})$.

*Necessity.* Let $h$ be a permutation equivariant and sign invariant function, then for any uncanonicalizable eigenvector $\boldsymbol{u}$ with $\boldsymbol{P}\boldsymbol{u} = -\boldsymbol{u}$, we have $h(\boldsymbol{P}\boldsymbol{u}) = h(-\boldsymbol{u}) = h(\boldsymbol{u})$. Following the proof of Corollary E.6, for any two permuted values $u_i$ and $u_j$ with $u_i = -u_j$, there exists a pairwise permutation matrix $\boldsymbol{P}$ such that $(\boldsymbol{P}\boldsymbol{u})_i = \boldsymbol{u}_j = -\boldsymbol{u}_i$. Further combined with $h(\boldsymbol{P}\boldsymbol{u}) = h(\boldsymbol{u})$, we have $h(\boldsymbol{u})_i = h(\boldsymbol{P}\boldsymbol{u})_i = (\boldsymbol{P}h(\boldsymbol{u}))_i = h(\boldsymbol{u})_j$. Thus, two opposite numbers in $\boldsymbol{u}$ always have the same output on their corresponding rows of $h(\boldsymbol{u})$.

Given this property, we can choose random opposite signs for equal numbers in $|\boldsymbol{u}|$ (*i.e.*, a flipping), denoted as $\tilde{\boldsymbol{u}} = \mathrm{rand\_flip}(|\boldsymbol{u}|)$. Let $\phi(|\boldsymbol{u}|) = h(\mathrm{rand\_flip}(|\boldsymbol{u}|))$. It is easy to verify that $\phi(|\boldsymbol{u}|)$ is well-defined as different flipping leads to the same output. Specifically, this is because:

1. Opposite numbers in $\boldsymbol{u}$ leads to the same output on their corresponding rows of $h(\boldsymbol{u})$;

2. $\tilde{\boldsymbol{u}}$'s produced by different flippings differ only by a permutation;

3. $h$ is permutation equivariant.

Thus numbers in $\boldsymbol{u}$ with the same absolute value lead to the same outputs, and these outputs are consistent for different flippings. $\qquad\square$

Then we prove Theorem 3.4.

*Proof.* We construct $\phi$ as follows. If $\mathbb{1}_{\boldsymbol{u}} = 1$, meaning $\boldsymbol{u}$ is canonicalizable, we let $\phi$ as in Lemma E.4; and if $\mathbb{1}_{\boldsymbol{u}} = 0$, meaning $\boldsymbol{u}$ is uncanonicalizable, we let $\phi$ as in Lemma E.7. Combining the sufficiency and necessity of these two lemmas gives the proof of Theorem 3.4. $\qquad\square$

## E.5 Proof of Corollary 3.5

Before proving Corollary 3.5, we first give some intuition about why SignNet is non-universal. In Theorem 3.4 we have already seen that, on uncanonicalizable eigenvectors, the inner $\phi$ function of SignNet is equal to some other function that only takes in the absolute value of the input eigenvector. This may not be a problem if we are only considering one eigenspace, since a permutation equivariant and sign invariant function is supposed to behave that way; there is no loss of information. The problem occurs when SignNet is applied to multiple eigenvectors, where each eigenvector is passed to the $\phi$ function separately before concatenated and fed to $\rho$. We can imagine that taking the absolute value of one uncanonicalizable eigenvector would lose some relative positional information about which elements are positive and which elements are negative, with respect to other eigenvectors. This loss of information makes SignNet non-universal.

For instance, we may try to construct a simple counterexample where taking the absolute value loses some information and as a result, SignNet fails to distinguish two non-isomorphic matrices. Drawing from the intuition above, here is our first try:

$$\boldsymbol{U}_1 = [\boldsymbol{u}_{11}, \boldsymbol{u}_{12}] = \begin{pmatrix} 1 & -1 & 1 & -1 \\ 2 & 3 & 4 & 5 \end{pmatrix}^\top,$$

$$\boldsymbol{U}_2 = [\boldsymbol{u}_{21}, \boldsymbol{u}_{22}] = \begin{pmatrix} -1 & 1 & 1 & -1 \\ 2 & 3 & 4 & 5 \end{pmatrix}^\top.$$

Suppose the first column eigenvector of $\boldsymbol{U}_1$ and $\boldsymbol{U}_2$ corresponds to eigenvalue $\lambda_1 = 1$, the second column eigenvector of $\boldsymbol{U}_1$ and $\boldsymbol{U}_2$ corresponds to eigenvalue $\lambda_2 = 2$, and other eigenvectors not shown corresponds to eigenvalue $0$ (so we safely ignore them). Then the Laplacian matrices corresponding to $\boldsymbol{U}_1$ and $\boldsymbol{U}_2$ are:

$$\boldsymbol{L}_1 = \lambda_1 \boldsymbol{u}_{11} \boldsymbol{u}_{11}^\top + \lambda_2 \boldsymbol{u}_{12} \boldsymbol{u}_{12}^\top = \begin{pmatrix} 9 & 11 & 17 & 19 \\ 11 & 19 & 23 & 31 \\ 17 & 23 & 33 & 39 \\ 19 & 31 & 39 & 51 \end{pmatrix},$$

$$\boldsymbol{L}_2 = \lambda_1 \boldsymbol{u}_{21} \boldsymbol{u}_{21}^\top + \lambda_2 \boldsymbol{u}_{22} \boldsymbol{u}_{22}^\top = \begin{pmatrix} 9 & 11 & 15 & 21 \\ 11 & 19 & 25 & 29 \\ 15 & 25 & 33 & 39 \\ 21 & 29 & 39 & 51 \end{pmatrix}.$$

$\boldsymbol{L}_1$ and $\boldsymbol{L}_2$ are clearly non-isomorphic, as there does not exist a permutation matrix $\boldsymbol{P}$ such that $\boldsymbol{L}_1 = \boldsymbol{P} \boldsymbol{L}_2 \boldsymbol{P}^\top$. However, the second eigenvector of $\boldsymbol{U}_1$ and $\boldsymbol{U}_2$ are the same; the first eigenvector of $\boldsymbol{U}_1$ and $\boldsymbol{U}_2$ are uncanonicalizable and according to Theorem 3.4, SignNet will treat them as the same. Therefore, SignNet fails to distinguish these two non-isomorphic matrices, and consequently it cannot approximate any permutation equivariant and sign invariant function that assigns different outputs to these matrices. However, as you may already notice, this example is not really legal, since the eigenvectors in $\boldsymbol{U}_1$ and $\boldsymbol{U}_2$ are not orthogonal (they are also not normalized, but whether normalizing or not does not affect our claims, so for simplicity we will just keep them unnormalized).

Next, we give a valid example with orthogonal eigenvectors, and show how SignNet fails to distinguish two non-isomorphic matrices in this case:

$$\boldsymbol{U}_1 = [\boldsymbol{u}_{11}, \boldsymbol{u}_{12}] = \begin{pmatrix} -1 & 1 & -1 & 1 & 2 & 2 & -2 & -2 & 0 & 0 \\ 1 & -1 & 1 & -1 & 1 & 1 & 0 & 0 & -1 & -1 \end{pmatrix}^\top,$$

$$\boldsymbol{U}_2 = [\boldsymbol{u}_{21}, \boldsymbol{u}_{22}] = \begin{pmatrix} 1 & 1 & -1 & -1 & 2 & 2 & -2 & -2 & 0 & 0 \\ 1 & -1 & -1 & 1 & 1 & -1 & 0 & 0 & -1 & 1 \end{pmatrix}^\top.$$

We still assume $\lambda_1 = 1$, $\lambda_2 = 2$. Their corresponding Laplacian matrices are:

$$
L_1 = \begin{pmatrix}
3 & -3 & 3 & -3 & 0 & 0 & 2 & 2 & -2 & -2 \\
-3 & 3 & -3 & 3 & 0 & 0 & -2 & -2 & 2 & 2 \\
3 & -3 & 3 & -3 & 0 & 0 & 2 & 2 & -2 & -2 \\
-3 & 3 & -3 & 3 & 0 & 0 & -2 & -2 & 2 & 2 \\
0 & 0 & 0 & 0 & 6 & 6 & -4 & -4 & -2 & -2 \\
0 & 0 & 0 & 0 & 6 & 6 & -4 & -4 & -2 & -2 \\
2 & -2 & 2 & -2 & -4 & -4 & 4 & 4 & 0 & 0 \\
2 & -2 & 2 & -2 & -4 & -4 & 4 & 4 & 0 & 0 \\
-2 & 2 & -2 & 2 & -2 & -2 & 0 & 0 & 2 & 2 \\
-2 & 2 & -2 & 2 & -2 & -2 & 0 & 0 & 2 & 2
\end{pmatrix},
$$

$$
L_2 = \begin{pmatrix}
3 & -1 & -3 & 1 & 4 & 0 & -2 & -2 & -2 & 2 \\
-1 & 3 & 1 & -3 & 0 & 4 & -2 & -2 & 2 & -2 \\
-3 & 1 & 3 & -1 & -4 & 0 & 2 & 2 & 2 & -2 \\
1 & -3 & -1 & 3 & 0 & -4 & 2 & 2 & -2 & 2 \\
4 & 0 & -4 & 0 & 6 & 2 & -4 & -4 & -2 & 2 \\
0 & 4 & 0 & -4 & 2 & 6 & -4 & -4 & 2 & -2 \\
-2 & -2 & 2 & 2 & -4 & -4 & 4 & 4 & 0 & 0 \\
-2 & -2 & 2 & 2 & -4 & -4 & 4 & 4 & 0 & 0 \\
-2 & 2 & 2 & -2 & -2 & 2 & 0 & 0 & 2 & -2 \\
2 & -2 & -2 & 2 & 2 & -2 & 0 & 0 & -2 & 2
\end{pmatrix}.
$$

$L_1$ and $L_2$ are non-isomorphic. In fact, there are 24 0's in $L_1$ but only 16 0's in $L_2$, so there is no way they are isomorphic. However, we can verify three things:

1. The eigenvectors are legal, *i.e.*, $u_{11}^\top u_{12} = u_{21}^\top u_{22}^\top = 0$;
2. All four eigenvectors $u_{11}, u_{12}, u_{21}, u_{22}$ are uncanonicalizable;
3. $u_{11}$ and $u_{21}$ have the same absolute value, $u_{12}$ and $u_{22}$ have the same absolute value.

Therefore, by Theorem 3.4, SignNet would give identical output for $U_1$ and $U_2$. Thus it cannot approximate permutation equivariant and sign invariant functions that assign different outputs to these matrices. Unconvinced readers could implement a randomly initialized SignNet themselves and verify our claim (we have already done that).

Next we discuss BasisNet. We first introduce BasisNet. Let $U_i \in \mathbb{R}^{n \times d_i}$ be an orthonormal basis of a $d_i$ dimensional eigenspace. Then BasisNet is parameterized by

$$f(U_1, \ldots, U_l) = \rho\left([\phi_{d_i}(U_i U_i^\top)]_{i=1}^l\right),$$

where each $\phi_{d_i}$ is shared amongst all subspaces of the same dimension $d_i$, and $l$ is the number of eigenspaces. Since higher-order permutation equivariant networks are intractable, here we mainly focus on using first-order permutation equivariant $\phi_{d_i}$. This means $\phi_{d_i}(PU_iU_i^\top P^\top) = P\phi_{d_i}(U_iU_i^\top)$ for all permutation matrix $P$.

As a special case, if $d_i = 1$, by letting $\phi(u) = \phi_1(uu^\top)$, BasisNet is reduced to SignNet:

$$f(u_1, \ldots, u_l) = \rho\left([\phi(u_i)]_{i=1}^l\right).$$

Since SignNet is not universal, there exists a permutation equivariant and sign invariant function acting on matrices whose eigenvalue multiplicity is no more than 1, and SignNet is not able to approximate it. Since BasisNet is reduced to SignNet when $d_i = 1$, BasisNet is also incapable of approximating this function. Therefore BasisNet is not universal either.

In fact Theorem 3.4 can be extended to basis, showing loss of information in higher dimension eigenspaces as well. By Ma et al. [36], eigenvectors $U \in \mathbb{R}^{n \times d}$ are uncanonicalizable under basis ambiguity *iff* there exists a permutation matrix $P$ such that $U \neq PU$ and $U = PUQ$ for some orthogonal matrix $Q$. We have the following theorem.

**Theorem E.8.** *If a function $h \colon \mathbb{R}^{n \times d} \to \mathbb{R}^{n \times d_{\text{out}}}$ is permutation equivariant and basis invariant on any uncanonicalizable eigenvectors $U$ with $U = PUQ$, then for any $i \neq j$ such that $U_i = U_jQ$, we have*

$$h(U)_i = h(U)_j.$$

*That is, the rows of $h(U)$ are constant on every orbit of the permutation $P$.*

*Proof.* Let $h\colon \mathbb{R}^{n\times d} \to \mathbb{R}^{n\times d_{\text{out}}}$ be a permutation equivariant and basis invariant function, and let $\boldsymbol{U} = \boldsymbol{P}\boldsymbol{U}\boldsymbol{Q}$ be any uncanonicalizable eigenvectors. For any $i \neq j$ such that $\boldsymbol{U}_i = \boldsymbol{U}_j\boldsymbol{Q}$, we claim that there exists a permutation matrix $\boldsymbol{P}'$ such that $\boldsymbol{U} = \boldsymbol{P}'\boldsymbol{U}\boldsymbol{Q}$ and $\boldsymbol{P}'$ permutes the $i$-th row to the $j$-th row. We prove this claim as follows.

Denote the permutation induced by $\boldsymbol{P}$ as $\boldsymbol{P}\colon i \mapsto \sigma(i)$, and the permutation induced by $\boldsymbol{P}'$ as $\boldsymbol{P}'\colon i \mapsto \sigma'(i)$. We wish to prove that there exists $\boldsymbol{P}'$ such that $\boldsymbol{U} = \boldsymbol{P}'\boldsymbol{U}\boldsymbol{Q}$ and $j = \sigma'(i)$. We construct $\boldsymbol{P}'$ as follows.

- If $\sigma(i) = j$, we could simply let $\boldsymbol{P}' = \boldsymbol{P}$.

- Otherwise, if $\sigma(i) = j' \neq j$, denote $j = \sigma(i')$ where $i' \neq i$. We let $\sigma'(i) = j$, $\sigma'(i') = j'$, and $\sigma'(k) = \sigma(k)$ for all other $k \neq i, i'$. Then clearly $\boldsymbol{U} = \boldsymbol{P}'\boldsymbol{U}\boldsymbol{Q}$ still holds.

By the claim above and observing $h(\boldsymbol{U}) = h(\boldsymbol{P}'\boldsymbol{U}\boldsymbol{Q}) = h(\boldsymbol{P}'\boldsymbol{U})$, we have

$$h(\boldsymbol{U})_j = h(\boldsymbol{P}'\boldsymbol{U})_j = \bigl(\boldsymbol{P}'h(\boldsymbol{U})\bigr)_j = h(\boldsymbol{U})_i.$$

$\square$

Theorem E.8 indicates loss of information when BasisNet is applied to multiple eigenspaces with uncanonicalizable eigenvectors.

## E.6 Proof of Theorem 4.1

*Proof.* It suffices to show that the projections $\{\mathscr{P}\boldsymbol{e}_i\}_{1\leq i\leq n}$ of the standard basis vectors $\{\boldsymbol{e}_i\}_{1\leq i\leq n}$ onto the eigenspace still has full rank. Then since the dimension of the eigenspace is $d$, we can select $d$ projection vectors such that they remain linearly independent.

This is easy to prove. For any vector $\boldsymbol{a} \in \mathbb{R}^n$, there exists a set of coefficients $a_1, \ldots, a_n$ such that $\boldsymbol{a} = a_1\boldsymbol{e}_1 + \cdots + a_n\boldsymbol{e}_n$. In particular, this holds for vectors in the eigenspace $V$. Thus multiplying both sides by $\mathscr{P}$ we have $\boldsymbol{a} = \mathscr{P}\boldsymbol{a} = a_1\mathscr{P}\boldsymbol{e}_1 + \cdots + a_n\mathscr{P}\boldsymbol{e}_n$ for all $\boldsymbol{a} \in V$. Thus the projections $\{\mathscr{P}\boldsymbol{e}_i\}_{1\leq i\leq n}$ are still a set of basis in the eigenspace, and Theorem 4.1 follows. $\square$

## E.7 Proof of Theorem 4.2

*Proof.* When permutation equivariance is not required, all eigenvectors are canonicalizable. Thus "optimal" here means Algorithm 1 and Algorithm 2 can canonicalize all eigenvectors. Algorithm 1 is clearly an optimal orbit canonicalization since:

1. It outputs either $\boldsymbol{u}$ or $-\boldsymbol{u}$ and is an orbit canonicalization;

2. Every eigenvector has at least one non-zero element, so Algorithm 1 always succeeds;

3. It is sign invariant, since $\boldsymbol{u}$ and $-\boldsymbol{u}$ gives the same output (*i.e.*, the one such that its first non-zero entry is positive).

Algorithm 2 is also an optimal orbit canonicalization since:

1. The Gram-Schmidt orthogonalization of a set of basis in a subspace is still a set of basis in that subspace (but orthogonalized), so Algorithm 2 is an orbit canonicalization;

2. By Theorem 4.1, Algorithm 2 always succeeds in finding the indices $i_1 < \cdots < i_d$ and orthogonalizing them;

3. It is basis invariant, since the projections $\{\mathscr{P}\boldsymbol{e}_i\}$ are basis invariant, and all procedures that follow remain basis invariant.

$\square$

### E.8 Proof of Theorem 4.3 and Corollary 4.4

We strongly recommend the readers to read Appendix B first to get some intuition about our algorithm. Before proving its superiority, we note that Algorithm 3 is an orbit canonicalization, and is basis invariant and permutation equivariant:

1. The Gram-Schmidt orthogonalization of a set of basis in a subspace is still a set of basis in that subspace (but orthogonalized), so Algorithm 3 is an orbit canonicalization;

2. It is basis invariant, since the projections $\{p_i\} = \{\mathscr{P}e_i\}$ are basis invariant, and all procedures that follow remain basis invariant;

3. The permutation equivariance of Algorithm 3 is more technical, which is proved in Theorem B.2 in Appendix B.1.

We prove the superiority of OAP-lap over MAP.

*Proof.* OAP is strictly superior to MAP in two aspects. Firstly, we notice that the computation of summary vectors $x_i$ in these two algorithms is almost the same, except with different values for $\alpha_i$. In OAP, the $\alpha_i$ is defined by $\alpha_i = \text{hash}(\mathscr{P}_{ii}, \{\!\{\mathscr{P}_{ij}\}\!\}_{j\neq i})$, while in MAP, their $\alpha_i' = \|\mathscr{P}_i\|$ is a special case of ours. Thus:

- If $\alpha_i = \alpha_j$, then $\mathscr{P}_{ii} = \mathscr{P}_{jj}$ and $\{\!\{\mathscr{P}_{ik}\}\!\} = \{\!\{\mathscr{P}_{jk}\}\!\}$. This indicates $\alpha_i' = \|\mathscr{P}_i\| = \|\mathscr{P}_j\| = \alpha_j'$.

- However, if $\alpha_i' = \alpha_j'$, this does not necessarily mean $\alpha_i = \alpha_j$. For instance, let $\mathscr{P}_1 = \mathscr{P}_2 = (1, 0)$, then $\alpha_1' = \alpha_2'$ but $\alpha_1 \neq \alpha_2$.

This means $\alpha_i$ is more "distinctive" than $\alpha_i'$, and as a result we will have a better chance finding indices $i_1 < \cdots < i_d$ such that $\mathscr{P}x_{i_1}, \ldots, \mathscr{P}x_{i_d}$ is a basis of the eigenspace. This is because, more "distinctive" $\alpha_i$ would lead to more "refined" (smaller) groups $\mathcal{B}_i$, and as a result, the rank of the projections $\{\mathscr{P}x_i\}$ will not decrease (and possibly increase). By splitting the group $\mathcal{B}_i$ into several smaller groups, the original summary vector can be expressed as a linear combination of all new summary vectors, thus the rank of projections of summary vectors does not decrease.

Secondly, OAP is superior to MAP by allowing flexible choices of $i_1 < \cdots < i_d$. In fact, after working out the summary vectors, the rest of the MAP algorithm is equivalent to Algorithm 3 except that they restrict $i_1 = 1, \ldots, i_d = d$. Recall that in MAP, given $u_1, \ldots, u_{i-1}$ already found, they look for $u_i$ in the orthogonal complementary space of $\langle u_1, \ldots, u_{i-1} \rangle$ in $V$ by projecting $x_i$ onto that space; and if $x_i$ is orthogonal to that space, their algorithm fails. This is exactly what Gram-Schmidt process does to the projection vectors $\{\mathscr{P}x_i\}$. Except in our algorithm, instead of looking at projections $\mathscr{P}x_1, \ldots, \mathscr{P}x_d$, we look for all projections $\mathscr{P}x_1, \ldots, \mathscr{P}x_k$ and try to select a subset of them such that the algorithm succeeds. This is obviously strictly superior, since there may be cases where $\mathscr{P}x_1, \ldots, \mathscr{P}x_d$ is not a basis of the eigenspace but another subset of $\mathscr{P}x_1, \ldots, \mathscr{P}x_k$ is. $\square$

Next we prove the superiority of OAP-graph over FA-graph.

FA-graph works as follows. Assume the graph has $k$ eigenspaces with projection matrices $\mathscr{P}_1, \ldots, \mathscr{P}_k$. Let $s_1, \ldots, s_k$ be the diagonals of these projection matrices, and let $S = [s_1, \ldots, s_k] \in \mathbb{R}^{n \times k}$. Then the frame of the graph is defined to be all permutations that sort the rows of $S$ in column lexicographic order, and the canonical form of the graph is the set of graphs produced by these permutations (by applying them to the original graph). We call this canonicalization FA-graph.

Next we introduce OAP-graph. Let $\alpha_i^\ell$ be defined in the same way as $\alpha_i$, except that $\ell = 1, \ldots, k$ denotes that $\alpha_i^\ell$ is computed from the projection matrix of the $\ell$-th eigenspace $\mathscr{P}_\ell$. Let $S' \in \mathbb{R}^{n \times k}$ be a matrix with $S_{i\ell} = \alpha_i^\ell$. The frame of the graph is the set of all permutations that sort the rows of $S'$ in column lexicographic order, and the canonicalization of the graph is the set of graphs produced by these permutations. We name this canonicalization OAP-graph. FA-graph in Puny et al. [45] works in a similar fashion except with different definition for $\alpha_i^\ell$. Since the $\alpha_i^\ell$ used in OAP-graph is stronger than in FA-graph, OAP-graph is also superior to FA-graph.

We will show that FA-graph is equivalent to our algorithm that takes $\alpha_i = \mathscr{P}_{ii}$, which is weaker than our definition $\alpha_i = \text{hash}(\mathscr{P}_{ii}, \{\!\!\{\mathscr{P}_{ij}\}\!\!\})$.

*Proof.* OAP-graph is almost identical to FA-graph, except that instead of sorting $\alpha_i' = \mathscr{P}_{ii}$ as in FA-graph, we sort values of $\alpha_i = \text{hash}(\mathscr{P}_{ii}, \{\!\!\{\mathscr{P}_{ij}\}\!\!\}_{j \neq i})$.

1. If $\alpha_i = \alpha_j$, clearly $\alpha_i' = \mathscr{P}_{ii} = \mathscr{P}_{jj} = \alpha_j'$.

2. However, if $\alpha_i' = \alpha_j'$, this does not necessarily mean $\alpha_i = \alpha_j$. For instance, let $\mathscr{P}_1 = (1, 1)$ and $\mathscr{P}_2 = (2, 1)$, then $\alpha_1' = \alpha_2'$ but $\alpha_1 \neq \alpha_2$.

This means $\alpha_i$ is more "distinctive" than $\alpha_i'$. We know that the frame obtained by FA-graph or OAP-graph consists of all permutations that sort the values of $\alpha_i$ in lexicographic order. Thus, a strictly more distinctive $\alpha_i$ will lead to strictly less repeating elements and strictly less number of permutations (*i.e.*, strictly smaller frame size). $\square$

Finally, we prove the superiority of OAP-lap over FA-lap.

*Proof.* FA-lap is almost identical to OAP-lap, except that it uses a less distinctive $\alpha_i = \mathscr{P}_{ii}$. When proving the superiority of OAP-graph over FA-graph, we proved that $\alpha_i$ used by FA-lap is strictly less distinctive than $\alpha_i$ used by OAP-lap, and when proving the superiority of OAP-lap over MAP we proved more distinctive $\alpha_i$ would lead to higher chance of finding indices $i_1 < \cdots < i_d$. Combining these facts gives the proof of the superiority of OAP-lap over FA-lap. $\square$

# F  Augmentations of the OAP algorithm

In this section, we present three variations of the OAP algorithm. The first two variants enhance the original OAP algorithm (Algorithm 5) through straightforward augmentations, while the third variant extends the OAP algorithm to render it trainable by neural networks.

## F.1  Incorporating node features

Recall in Algorithm 5 we defined the permutation-equivariant vector $\boldsymbol{\alpha}$ with

$$\alpha_i = \text{hash}(\mathscr{P}_{ii}, \{\!\!\{\mathscr{P}_{ij}\}\!\!\}_{j \neq i}), \ 1 \leq i \leq n,$$

and used the values of $\alpha_i$ to split the standard basis vectors into different groups. From the proof of Theorem B.2, we can see that the permutation equivariance of OAP is guaranteed by the permutation equivariance of $\boldsymbol{\alpha}$, which is respected as long as the indices of $\alpha_i$ are permutation equivariant, *i.e.*,

$$\alpha_{\sigma(i)} = \text{hash}(\mathscr{P}_{\sigma(i)\sigma(i)}, \{\!\!\{\mathscr{P}_{\sigma(i)\sigma(j)}\}\!\!\}_{j \neq i}), \ 1 \leq i \leq n$$

for all permutation $\sigma$. This indicates that we can incorporate any permutation-equivariant features into $\alpha_i$, as long as it preserves the permutation equivariance of $i$. If the input graph has node features $\boldsymbol{X} \in \mathbb{R}^{n \times d}$, where $n$ is the number of nodes, then the node features $\boldsymbol{X}$ would be permutation equivariant. Using this property, we propose the following variant of OAP, by incorporating $\boldsymbol{X}$ into $\alpha_i$:

$$\alpha_i = \text{hash}(\mathscr{P}_{ii}, \{\!\!\{\mathscr{P}_{ij}\}\!\!\}_{j \neq i}, \boldsymbol{X}_i), \ 1 \leq i \leq n,$$

where $\boldsymbol{X}_i$ is the node feature of the $i$-th node. It is easy to see that this variant is superior to the original OAP algorithm.

## F.2  Incorporating information from other eigenvectors

Another example of permutation equivariant feature is the projection matrices of other eigenspaces. Suppose there are $\ell$ eigenspaces in total, with corresponding projection matrices $\mathscr{P}^{(1)}, \ldots, \mathscr{P}^{(\ell)}$. For each $1 \leq i \leq n$, we can calculate a value of $\alpha_i$ for each of these eigenspaces:

$$\alpha_i^{(k)} = \text{hash}(\mathscr{P}_{ii}^{(k)}, \{\!\!\{\mathscr{P}_{ij}^{(k)}\}\!\!\}_{j \neq i}), \ 1 \leq k \leq \ell,$$

then coalesce them into a single value

$$\alpha_i = \text{hash}(\alpha_i^{(s)}, \{\!\{\alpha_i^{(j)}\}\!\}_{j \neq s}),$$

where $s$ is the index of the eigenspace that we are trying to canonicalize. In this way, we can canonicalize eigenvectors from one eigenspace using information from all $\ell$ eigenspaces, instead of this eigenspace alone. Clearly this variant is also superior to the original OAP canonicalization.

### F.3 Learnable canonicalization

We can also adopt learnable networks to substitute certain components of the OAP canonicalization. For example, the hash function in the definition of $\alpha_i$ can be made learnable:

$$\alpha_i = h(\mathscr{P}_{ii}, \{\!\{\mathscr{P}_{ij}\}\!\}_{j \neq i}), \ 1 \leq i \leq n,$$

where $h$ is a neural network that is invariant to the ordering of $\{\!\{\mathscr{P}_{ij}\}\!\}_{j \neq i}$. We can also replace the Gram-Schmidt orthogonalization with a learnable network:

$$\boldsymbol{U}^* = h'(\mathscr{P}\boldsymbol{x}_{i_1}, \ldots, \mathscr{P}\boldsymbol{x}_{i_d}),$$

where $h'$ is a permutation-equivariant neural network. The output vectors may not be orthogonal, but since $h'$ almost certainly preserves the rank of $(\mathscr{P}\boldsymbol{x}_{i_1}, \ldots, \mathscr{P}\boldsymbol{x}_{i_d})$, it does not lose expressive power.

## G Related work

### G.1 Equivariant learning

There are four main approaches in the mainstream to achieve equivariance. The first approach, as used in Yarotsky [69], Murphy et al. [40], is group averaging. This approach involves averaging over the entire group, which becomes impractical when dealing with large groups. For example, the permutation group on graphs and the sign transformation group on single eigenvectors have an exponential number of elements, while the orthogonal transformation group has an infinite number of elements. In such cases, random sampling becomes necessary. For instance, relational pooling can be used for the permutation group [40], or random sign sampling can be used for the sign transformation group [14].

The second approach is frame averaging, which was proposed by Puny et al. [45]. This method involves averaging over a smaller frame instead of the entire group. However, a drawback of this approach is that it requires the frame to be equivariant, which is a strong assumption and often difficult to fulfill. Duval et al. [13] also suggested stochastic frame averaging by randomly sampling from the frame. They applied this method to a frame of size 8, given by the set $\{1, -1\}^3$.

The third approach is learned canonicalization, proposed by Kaba et al. [26], which can be viewed as frame averaging with a frame size of 1. However, such canonicalization is not possible if the input has non-trivial automorphism. Additionally, even for non-automorphic inputs, the assumption of equivariance makes constructing such canonicalization challenging.

The fourth approach involves designing equivariant network architectures tailored to specific equivariant constraints. For example, Fuchs et al. [18] introduced the SE(3)-transformer, which exhibits equivariance to the SE(3) group. Additionally, Zaheer et al. [70] and Segol and Lipman [53] proposed networks that are permutation invariant and equivariant. Furthermore, Lim et al. [33] and Lim et al. [32] presented networks that maintain sign invariance, basis invariance, and sign equivariance. However, this approach's downside lies in the increased complexity and training cost associated with enforcing equivariance at the network level. Moreover, its applicability is limited to specific groups and cannot be extended to arbitrary ones.

In our paper, we introduce a different theoretical framework for canonicalization that unifies previous approaches while achieving the highest efficiency.

### G.2 Canonicalization

There have been several related works using canonicalization to achieve equivariance. Apart from Kaba et al. [26] discussed in Section 2.1, Dym et al. [16] proved the impossibility of continuity for

canonicalizations with single canonical forms, which explains the subpar performance of some current canonization methods. Since in our framework, the canonical form is defined more generally as a set rather than a single element, it allows for more flexibility and continuity with weighted averaging. Lin et al. [34] proposed a framework of canonicalization that assumes a single canonical form, so their framework is essentially a special case of our framework by taking $|\mathcal{C}(X)| = 1$. The "minimal frame" in their paper is just the induced frame of an optimal canonicalization in our framework. However, the assumption of a single canonical form has the following key limitations:

- *Theoretical Impossibility.* A single canonical form may be theoretically impossible under some constraints. For instance, canonicalization of LapPE requires permutation equivariance, in this case it is theoretically impossible to find a single canonical form for some eigenvectors [36].

- *Computational Intractability.* A single canonical form may be computationally intractable to find. For instance, computing graph canonicalization is NP-hard [5]. In this case, we have to adopt an approximate approach that admits a set of canonical forms, as in our EXP experiment in Section 5.1.

Even if we could find a single canonical form for all inputs, as pointed out by Dym et al. [16] such canonicalizations are still not continuous, which hurts their performance.

In summary, compared to previous canonicalization works that only consider single-element canonicalization that may be unrealizable, we take canonicalizability into account throughout the analysis and thus gives a more rigorous and general framework that works for both canonicalizable (single canonical form) and uncanonicalizable (no single canonical form) inputs.

### G.3 Canonical forms

Canonical forms, a extensively studied topic in mathematics [41], have found applications in various domains, including machine learning. While limited theoretical discussions have been conducted regarding its application in equivariant learning, several practical implementations of canonical forms have been proposed in prior research. For example, ConDor, a self-supervised method introduced by Sajnani et al. [51], focuses on learning to canonicalize the 3D orientation and position of 3D point clouds. Another approach, known as Canonical Field Network (CaFi-Net), was presented by Agaram et al. [2], employing self-supervision to canonicalize the 3D pose of instances represented by neural fields, such as neural radiance fields (NeRFs). Additionally, Sun et al. [56] proposed an unsupervised capsule architecture specifically designed for 3D point clouds. Furthermore, Ma et al. [35] uses canonicalization to achieve state and action invariance in generative models. In this paper, we contribute rigorous theoretical foundations for canonicalization, specifically addressing the eigenvector canonicalization problem, which holds significant practical implications.

### G.4 Sign invariance

Several approaches have been proposed in prior research to tackle the issue of sign ambiguity. One commonly used heuristic is Random Sign (RS), which randomly flips the signs of eigenvectors during training to promote insensitivity to different signs [14]. However, this approach still requires the network to learn these signs, resulting in slower convergence and a more challenging curve fitting task. Another method developed by Bro et al. [10] is a data-dependent approach that selects signs for each singular vector of a singular value decomposition. Nevertheless, in the worst case scenario, the chosen signs can be arbitrary, and this method does not handle rotational ambiguities in higher dimensional eigenspaces. Additionally, Kreuzer et al. [29] introduced the use of relative Laplacian embeddings of two nodes, but their approach encounters a significant computational bottleneck with a complexity of $\mathcal{O}(n^4)$. Another proposal, known as SignNet [33], feeds both $\boldsymbol{u}$ and $-\boldsymbol{u}$ to the same network, combines the outputs, and then passes them to another network. However, since the outputs for both $\boldsymbol{u}$ and $-\boldsymbol{u}$ need to be computed, this approach increases the training cost. Furthermore, Ma et al. [36] proposed MAP canonicalization to address the ambiguities of Laplacian eigenvectors during pre-processing. However, their method cannot be applied to general eigenvectors.

### G.5 Basis invariance

Lim et al. [33] introduced BasisNet, which is basis invariant. However, achieving universality with BasisNet necessitates utilizing higher-order tensors in $\mathbb{R}^{n^k}$, where $k$ can reach values as high as $n$ [37, 49], making BasisNet impractical. Huang et al. [23] extended BasisNet with SPE, which is both basis-invariant and resistant to perturbations in the Laplacian matrix. However, SPE still encounters exponential complexity akin to BasisNet. Zhang et al. [74] theoretically discussed the expressive power of spectral invariant networks, which include BasisNet. While Ma et al. [36] proposed the MAP canonicalization to tackle ambiguities in Laplacian eigenvectors during pre-processing, their approach is limited to specific eigenvectors, and their canonicalization on basis is relatively weak. In our work, we propose the OAP canonicalization, which is strictly superior to previous methods while incurring negligible computational overhead.

Beyond GNN, there exists literature exploring Laplacian eigenvectors of graphs and addressing the sign/basis ambiguity issues. Lai and Zhao [31] presented a solution employing optimal transport theory to tackle sign/basis ambiguities, involving the resolution of a non-convex optimization problem. However, this approach may be less efficient compared to canonicalization methods. In a similar vein, Tam and Dunson [58] proposed symmetrizing the embedding using a heuristic measure called ELD, which bears resemblance to the structure of SignNet.

## H   Dataset details

EXP [1] (GPL-3.0 License) is a dataset designed to explicitly evaluate the expressiveness of GNN models, and consists of a set of graph instances $\{\mathcal{G}_1, \ldots, \mathcal{G}_n, \mathcal{H}_1, \ldots, \mathcal{H}_n\}$, such that each instance is a graph encoding of a propositional formula. The classification task is to determine whether the formula is satisfiable (SAT). Each pair $(\mathcal{G}_i, \mathcal{H}_i)$ respects the following properties:

1. $\mathcal{G}_i$ and $\mathcal{H}_i$ are non-isomorphic;
2. $\mathcal{G}_i$ and $\mathcal{H}_i$ have different SAT outcomes, that is, $\mathcal{G}_i$ encodes a satisfiable formula, while $\mathcal{H}_i$ encodes an unsatisfiable formula;
3. $\mathcal{G}_i$ and $\mathcal{H}_i$ are 1-WL indistinguishable, so are guaranteed to be classified in the same way by standard MPNNs;
4. $\mathcal{G}_i$ and $\mathcal{H}_i$ are 2-WL distinguishable, so can be classified differently by higher-order GNNs.

The $n$-body problem dataset [28, 18] (MIT License) consists of a collection of $n = 5$ particles systems. Five particles each carry either a positive or a negative charge and exert repulsive or attractive forces on each other. The input to the network is the position of a particle in a specific time step, its velocity, and its charge. The task of the algorithm is then to predict the relative location and velocity 500 time steps into the future. The original problem is conducted in dimension $d = 3$. To test the scalability of different methods we also extend this experiment to higher dimensions $d = 5, 7, 9, 11$, as practiced in Lim et al. [32].

ZINC [24] (MIT License) consists of 12K molecular graphs from the ZINC database of commercially available chemical compounds. These molecular graphs are between 9 and 37 nodes large. Each node represents a heavy atom (28 possible atom types) and each edge represents a bond (3 possible types). The task is to regress constrained solubility (logP) of the molecule. The dataset comes with a predefined 10K/1K/1K train/validation/test split.

OGBG-MOLTOX21 and OGBG-MOLPCBA [22] (MIT License) are molecular property prediction datasets adopted by OGB from MoleculeNet. These datasets use a common node (atom) and edge (bond) featurization that represent chemophysical properties. OGBG-MOLTOX21 is a multi-mask binary graph classification dataset where a qualitative (active/inactive) binary label is predicted against 12 different toxicity measurements for each molecular graph. OGBG-MOLPCBA is also a multi-task binary graph classification dataset from OGB where an active/inactive binary label is predicted for 128 bioassays.

## I   Hyper-parameter settings

All (preliminary, failed and main) experiments are run on NVIDIA 3090 GPUs with 24GB memory.

## I.1 EXP experiment

In the EXP experiment, the hyper-parameters are not tuned. We use the same hyper-parameters as in Puny et al. [45]. Additionally, we also use the code base on EXP to test FA-GIN+ID and FA-GINE+ID on the ZINC dataset. For both models we use a 16-layer GNN, with the hidden dimension set to 176. The output of the GNN is then fed to a 2-layer MLP to produce the final regression output. On both EXP and ZINC the frame or canonicalization of graphs are computed during pre-processing time (using the `pre_transform` argument of the dataset class). On EXP, since the frame or canonicalization size for some graphs is large shown in Table 2 (because the EXP dataset is highly symmetric), we randomly sample from the frame or canonicalization, with the sampling size being 64. All experiments results are averaged over 4 runs with different random seeds.

## I.2 $n$-body experiment

In the $n$-body experiment, we tune the number of layers $L$ and the hidden dimension $h$ of our canonical averaging models. Other hyper-parameters such as learning rate are kept the same with Puny et al. [45]. The hyper-parameter settings for different methods with dimension $d = 3$ are reported in Table 8. For higher dimensions, we slightly adjust the hidden dimension $h$ ($\pm 3$) such that $h$ is a multiple of $d$.

Table 8: Hyper-parameter settings of different methods in the $n$-body experiment with dimension $d = 3$.

| Method | $L$ | $h$ | #param |
|---|---|---|---|
| Frame Averaging | 4 | 60 | 92103 |
| Sign Equivariant | 5 | 45 | 201555 |
| OAP-eig | 4 | 63 | 117817 |
| OAP-lap | 4 | 63 | 117817 |

## I.3 ZINC experiment

In the ZINC graph regression experiment, We follow the same settings as in Dwivedi et al. [15] for models with no PE, LapPE or LSPE, and the same settings as in Lim et al. [33] for models with SignNet, MAP or OAP. We implement FA-GIN+ID and FA-GINE+ID on ZINC by ourselves (their hyper-parameters are already reported above); all other baseline scores reported in Table 4 are taken from the original papers. The main hyper-parameters in our experiments are listed as follows.

- $k$: the number of eigenvectors used in the PE.
- $L_1$: the number of layers of the base model.
- $h_1$: the hidden dimension of the base model.
- $h_2$: the output dimension of the base model.
- $\lambda$: the initial learning rate.
- $t$: the patience of the learning rate scheduler.
- $r$: the factor of the learning rate scheduler.
- $\lambda_{\min}$: the minimum learning rate of the learning rate scheduler.
- $L_2$: the number of layers of SignNet or the normal GNN[6] (when using canonicalization as PE).
- $h_3$: the hidden dimension of SignNet or the normal GNN (when using canonicalization as PE).

The values of these hyper-parameters in our experiments are listed in Table 9.

## I.4 OGBG experiment

In the OGBG-MOLTOX21 experiment, the values of these hyper-parameters are listed in Table 10.

---

[6]We substitute SignNet with a normal GNN (Masked GIN) when using canonicalization as the PE method.

Table 9: Hyper-parameter settings of different models with different PE methods on ZINC.

| Model | PE | $k$ | $L_1$ | $h_1$ | $h_2$ | $\lambda$ | $t$ | $r$ | $\lambda_{\min}$ | $L_2$ | $h_3$ |
|---|---|---|---|---|---|---|---|---|---|---|---|
| GatedGCN | None | 0 | 16 | 78 | 78 | 0.001 | 25 | 0.5 | 1e-6 | - | - |
| | LapPE + RS | 8 | 16 | 78 | 78 | 0.001 | 25 | 0.5 | 1e-6 | - | - |
| | SignNet | 8 | 16 | 67 | 67 | 0.001 | 25 | 0.5 | 1e-6 | 8 | 67 |
| | MAP | 8 | 16 | 69 | 67 | 0.001 | 25 | 0.5 | 1e-5 | 6 | 69 |
| | OAP | 8 | 16 | 68 | 68 | 0.001 | 25 | 0.5 | 1e-5 | 6 | 68 |
| | OAP + LSPE | 8 | 13 | 59 | 59 | 0.001 | 25 | 0.5 | 1e-5 | 6 | 59 |
| PNA | None | 0 | 16 | 70 | 70 | 0.001 | 25 | 0.5 | 1e-6 | - | - |
| | LapPE + RS | 8 | 16 | 80 | 80 | 0.001 | 25 | 0.5 | 1e-6 | - | - |
| | SignNet | 8 | 16 | 70 | 70 | 0.001 | 25 | 0.5 | 1e-6 | 8 | 70 |
| | MAP | 8 | 16 | 70 | 70 | 0.001 | 25 | 0.5 | 1e-6 | 6 | 70 |
| | OAP | 8 | 16 | 70 | 70 | 0.001 | 25 | 0.5 | 1e-6 | 6 | 70 |
| | OAP + LSPE | 8 | 13 | 60 | 60 | 0.001 | 25 | 0.5 | 1e-6 | 6 | 60 |

Table 10: Hyper-parameter settings of different models with different PE methods on MOLTOX21.

| Model | PE | $k$ | $L_1$ | $h_1$ | $h_2$ | $\lambda$ | $t$ | $r$ | $\lambda_{\min}$ | $L_2$ | $h_3$ |
|---|---|---|---|---|---|---|---|---|---|---|---|
| GatedGCN | None | 0 | 8 | 154 | 154 | 0.001 | 25 | 0.5 | 1e-5 | - | - |
| | LapPE + RS | 3 | 8 | 154 | 154 | 0.001 | 25 | 0.5 | 1e-5 | - | - |
| | SignNet* | 3 | 8 | 150 | 150 | 0.001 | 22 | 0.14 | 1e-5 | 6 | 150 |
| | MAP | 3 | 8 | 150 | 150 | 0.001 | 22 | 0.14 | 1e-5 | 8 | 150 |
| | OAP | 3 | 8 | 160 | 150 | 0.001 | 22 | 0.14 | 1e-5 | 6 | 160 |
| PNA | None | 0 | 8 | 206 | 206 | 0.0005 | 10 | 0.8 | 2e-5 | - | - |
| | LapPE + RS | 16 | 8 | 140 | 140 | 0.0005 | 10 | 0.8 | 2e-5 | - | - |
| | SignNet* | 16 | 8 | 110 | 110 | 0.0005 | 10 | 0.8 | 8e-5 | 6 | 110 |
| | MAP | 16 | 8 | 115 | 113 | 0.0005 | 10 | 0.8 | 8e-5 | 7 | 115 |
| | OAP | 16 | 8 | 115 | 113 | 0.0005 | 10 | 0.8 | 8e-5 | 7 | 115 |

In the OGBG-MOLPCBA experiment, the values of these hyper-parameters are listed in Table 11.

Table 11: Hyper-parameter settings of different models with different PE methods on MOLPCBA.

| Model | PE | $k$ | $L_1$ | $h_1$ | $h_2$ | $\lambda$ | $t$ | $r$ | $\lambda_{\min}$ | $L_2$ | $h_3$ |
|---|---|---|---|---|---|---|---|---|---|---|---|
| GatedGCN | None | 0 | 8 | 154 | 154 | 0.001 | 25 | 0.5 | 1e-4 | - | - |
| | LapPE + RS | 3 | 8 | 154 | 154 | 0.001 | 25 | 0.5 | 1e-4 | - | - |
| | SignNet* | 3 | 8 | 200 | 200 | 0.001 | 25 | 0.5 | 1e-5 | 6 | 200 |
| | MAP | 3 | 8 | 200 | 200 | 0.001 | 25 | 0.5 | 1e-5 | 8 | 200 |
| | OAP | 3 | 8 | 200 | 200 | 0.001 | 25 | 0.5 | 1e-5 | 8 | 200 |
| PNA | None | 0 | 12 | 600 | 600 | 0.0005 | 10 | 0.8 | 2e-5 | - | - |
| | LapPE + RS | 16 | 12 | 600 | 600 | 0.0005 | 10 | 0.8 | 2e-5 | - | - |
| | SignNet* | 16 | 4 | 300 | 300 | 0.0005 | 10 | 0.8 | 2e-5 | 8 | 300 |
| | MAP | 16 | 4 | 304 | 304 | 0.0005 | 10 | 0.8 | 2e-5 | 8 | 304 |
| | OAP | 16 | 4 | 304 | 304 | 0.0005 | 10 | 0.8 | 2e-5 | 8 | 304 |

