# OpenReview forum: "A Canonicalization Perspective on Invariant and Equivariant Learning"
_NeurIPS.cc/2024/Conference — NeurIPS 2024 poster_

### Official Review · Reviewer_8tUB · 2024-07-11

**Soundness:** 3
**Presentation:** 2
**Contribution:** 3
**Rating:** 6
**Confidence:** 3

**Summary:**

The paper introduces a canonization perspective for designing frames in neural networks. Canonization maps inputs to their canonical forms, allowing efficient and even optimal frame design. The paper shows the connection between frames and canonical forms, leading to the development of new frames for eigenvectors under Orthogonal Axis Projection (OAP) that outperform existing methods. The reduction to canonization uncovers equivalences between previous methods and unifies existing invariant and equivariant frame averaging approaches.

**Strengths:**

- Theoretically describing frame through canonization is beneficial. Understanding the equivalence between frame and canonization (Theorem 3.1) is great, and the theoretical definition of optimal canonization (Theorem 3.2) is straightforward.
- The primary method, OAP, uses Gram-Schmidt orthogonalization to canonicalize eigenvectors and resolve eigenvalue degeneracy (or basis ambiguity), which is also straightforward.

**Weaknesses:**

### **Major**:
The overall writing could be improved. The authors claim to present a unified and essential view of equivariance learning, but the application is limited to the orthogonal group and eigenvector canonization. I would expect more examples to justify the theorem, such as the canonization view on the translation group or the product group $E(d)\times S_n$, similar to what frame averaging [1] achieves.


### **Minor**:
- The OAP results appear to be marginal, though considering this as a theoretical paper compensates for this shortcoming. I would like to know what is the result of MAP + LSPE in ZINC 500k compared to OAP + LSPE.
- Please add references in the main text to the appendix for proofs and n-body experiments to enhance readability.
- I suggest the authors add related works including [2] and [3]. I also hope the authors can give a discussion about the highly related concurrent works [4] and [5] to strengthen the contribution, not necessarily in this rebuttal since they are both new papers accepted in this ICML 2024, but please do consider it after this rebuttal.

[1] *Frame Averaging for Invariant and Equivariant Network Design*. Omri Puny, et al. ICLR 2022.

[2] *Learning Probabilistic Symmetrization for Architecture Agnostic Equivariance*. Jinwoo Kim, et al. NeurIPS 2023.

[3] *A Hitchhiker’s Guide to Geometric GNNs for 3D Atomic Systems*. Alexandre Duval, et al.

[4] *Equivariant Frames and the Impossibility of Continuous Canonicalization*. Nadav Dym, et al. ICML 2024.

[5] *Equivariance via Minimal Frame Averaging for More Symmetries and Efficiency*. Yuchao Lin, et al. ICML 2024.

**Questions:**

In line 992 (Appendix E.6), why is $\\{Pe_i\\}_{1≤i≤n}$ full rank? Shouldn't it have a rank of $\text{min}(n,d)$?

**Limitations:**

The authors do mention one of the limitations is that OAP does not fully resolve the optimality of eigenvector canonization under permutation equivariance.

---

> ### Author Rebuttal · Authors · 2024-08-06
>
> We thank Reviewer 8tUB for the constructive review and acknowledging our theoretical contributions. We address your main concerns as follows.
>
> ---
>
> **Q1.** The overall writing could be improved. The authors claim to present a unified and essential view of equivariance learning, but the application is limited to the orthogonal group and eigenvector canonization. I would expect more examples to justify the theorem, such as the canonization view on the translation group or the product group $E(d)\times S_n$, similar to what frame averaging [1] achieves.
>
> **A1.** Thanks for the suggestions. We will keep improving the writing to be more clear and readable.
>
> For the application of canonization, we provided multiple application scenarios of canonization in the paper:
>
> - **sign and basis equivariance**: we design optimal canonization algorithms (OAP) for sign and basis equviariance of eigenvectors, and show clear benefits in efficiency and performance.
> - **permutation equivariance**: in the EXP experiment (Section 5.1), we use canonization to achieve permutation equivariance on graph data (not sign/basis invariance of eigenvectors).
> - **the product group of the permutation group and the sign/basis transformation group:** for the graph case, we do consider the product group and show a strictly better canonization design (OAP for graphs).
> - **rotation equivariance:** In Appendix D, we applied canonization to achieve rotational equivariance of particles in PCA-frame methods.
>
> Besides, it is also easy to extend our results to the **translation group**. For example, a translation that moves the center of the input to the origin is a canonization under translation. Combining this canonization with the orthogonal group gives a canonization under the Euclidean group.
>
> In summary, we believe that these evidences show that our method has the generality to be applied to various tasks and different symmetries with improved efficiency over frame averaging. We will add these elaborations in the revision following your advice.
>
> ---
>
> **Q2.** I would like to know what is the result of MAP + LSPE in ZINC 500k compared to OAP + LSPE.
>
> **A2.** Following your suggestion, we further evaluate MAP + LSPE on ZINC. As shown in the following table, OAP can still outperform MAP under similar parameter constraints.
>
> | Model | #Param | MSE |
> | --- | --- | --- |
> | GatedGCN + MAP + LSPE | 475K | 0.101 ± 0.001 |
> | GatedGCN + OAP + LSPE | 491K | 0.098 ± 0.0009 |
> | PNA + MAP + LSPE | 550K | 0.104 ± 0.005 |
> | PNA + OAP + LSPE | 549K | 0.095 ± 0.004 |
>
> ---
>
> **Q3.** Please add references in the main text to the appendix for proofs and n-body experiments to enhance readability.
>
> **A3.** Thanks for the suggestion. We will provide these references in our revision.
>
> ---
>
> **Q4.** I suggest the authors add related works including [2] and [3]. I also hope the authors can give a discussion about the highly related concurrent works [4] and [5] to strengthen the contribution, not necessarily in this rebuttal since they are both new papers accepted in this ICML 2024, but please do consider it after this rebuttal.
>
> **A4.** Thanks for pointing out these related works! They are definitely related to the topic and we will discuss them in the revision. Among them, Kim et al [2] proposed a probabilistic frame averaging to achieve equivariance. Duval et al [3] gave a comprehensive review of geometric GNNs for 3D atomic systems, and introduces a taxonomy of geometric GNNs. We believe canonization methods can serve as a new direction for achieving the required equivariance of 3D atomic systems.
>
> More relevantly, Dym et al [4] proved the impossibility of continuity for canonizations with single canonical forms, which explains the subpar performance of some current canonization methods. Since in our framework, the canonical form is defined more generally as a set rather than a single element, it allows for more flexibility and continuity with weighted averaging. Lin et al [5] proposed a framework of canonization that assumes a single canonical form, so their framework is essentially a special case of our framework by taking $|\mathcal{C}(X)|=1$. The “minimal frame” in their paper is just the induced frame of an optimal canonization in our framework. However, the assumption of a single canonical form has the following key limitations:
>
> 1. **Theoretical Impossibility.** A single canonical form may be theoretically impossible under some constraints. For instance, canonization of LapPE requires permutation equivariance, in this case it is theoretically impossible to find a single canonical form for some eigenvectors.
> 2. **Computational Intractability.** A single canonical form may be computationally intractable to find. For instance, computing graph canonization is NP-hard. In this case, we have to adopt an approximate approach that admits a set of canonical forms, as in our EXP experiment.
>
> Even if we could find a single canonical form for all inputs, as pointed out by [4] such canonizations are still not continous, which hurts their performance.
>
> In summary, compared to previous canonization works that only consider single-element canonization that may be unrealizable, we take canonizability into account throughout the analysis and thus gives a more rigorous and general framework that works for both canonizable (single canonical form) and uncanonizable (no single canonical form) inputs.
>
> We will add these discussions in our paper.
>
> ---
>
> **Q5.** In line 992 (Appendix E.6), why is $\\{Pe_i\\}_{1\leq i\leq n}$ full rank? Shouldn't it have a rank of $\min(n,d)$?
>
> **A5.** In our paper $d$ refers to the dimension of the eigenspace, so we must have $d\leq n$. We will clarify this point in the proof.
>
> ---
>
> We thank Reviewer 8tUB again for the valuable review. We hope our response addresses your concerns. If you have further concerns, we are happy to address them in the discussion period.

---

> > ### Comment · Reviewer_8tUB · 2024-08-12
> > **Official Comment by Reviewer 8tUB**
> >
> > Thank you for your responses and I am satisfied with them. Please do include these discussions in the paper in future revisions. I have changed the score accordingly.

---

### Official Review · Reviewer_TcMZ · 2024-07-12

**Soundness:** 3
**Presentation:** 1
**Contribution:** 2
**Rating:** 5
**Confidence:** 3

**Summary:**

This paper highlights a one-to-one connection between frames over finite groups with ‘canonization’ over the space on which the group acts. This allows authors to prove non-universality of SignNets. Furthermore, authors claim that this view helps highlight equivalence of certain existing algorithms (MAP and FA-lap) arising in sign invariance and permutation equivariance problems, and ultimately leads to a proposal of a novel algorithm OAP.

**Strengths:**

Given the recent rise in papers covering the topic of frame averaging, this paper proposes a more useful view, by considering ‘canonizations’ instead, which are always smaller or same in size, compared to the corresponding frames. The proposed view turns out to be useful for proving universality under sign invariance and highlights issues with existing approaches for permutation equivariance - this makes it interesting for investigating in the setting of other groups.

The appendix seems to be quite informative and well-written.

**Weaknesses:**

This paper reads as if it was extremely rushed, resulting in a completely unreadable manuscript. There are a number of places where terms are not defined, the problem settings are not stated at all, and connections to existing work are not stated/clarified. Theorems are almost always presented without stating the full problem and without a clear outline of the assumptions. This is particularly true for superiority proofs, which clearly assume something about the hash function, but nothing is ever stated in regards to it. In addition to that, it is not immediately clear where the ‘canonization’ view can be useful, beyond the example provided in this paper.

Unfortunately, I am unable to comment on the numerical section, as I am not familiar with these experiments. However, given the problems in the rest of the manuscript, I urge the other reviewers to have a closer look at the numerical evaluation.

**Questions:**

Below I provide a number of comments and questions.

* First and foremost, while I realize the authors must be approaching from the graph side of the literature - within frame averaging the term used in all papers is ‘canonicalization’, and not ’canonization’, and I would suggest for consistency to refer to it as such, to avoid creating confusion.

* In subsection 2.1, as soon as frame averaging is presented, an assumption is made about finiteness of the group - but is never stated. Furthermore, after section 3.2, the vector spaces considered becomes simply finite dimensional Euclidean spaces.

* Line 87 and 95 - universal approximation is claimed, however neither a citation, nor a theorem reference is provided.

* In 110 you refer to LapPE. Ideally after introducing a term, you would directly provide a reference to it, instead of at the end of the next sentence.

* Line 128 - why are automorphisms of the input a major obstacle in analyzing complexity of frames?

* Line 144 - what you refer to as contractive ‘canonization’, is usually referred to as orbit canonicalization. Eg see  https://arxiv.org/abs/2402.16077

* Line 175 - the same thing is stated twice.

* In 3.3 and onwards, MAP is mentioned, however, since the work is built on top of it, an introduction to MAP (or at least an outline of what it is) should be present in the paper.

* In line 205 what is the norm of a vector?

* Whenever a theorem is stated, a reference to where the proof can be found should be stated.

* In both section 3 and 4 the actual problem being solved is never stated, making it very difficult to follow. By this I mean that given a matrix, you find its eigenvectors, and wish to construct a function which is permutation equivariant …

* Line 262 - what is the hash function? I can not seem to find any information about what it is, not in numerical experiments either, could you point to where this information is located?

* You mention that you have not been able to show that OAP is optimal - given the indices may not exist, how could it be optimal?

* Line 304 - you never say what FA/CA is - I presume this is frame averaging and canonisation averaging?

* Whenever superiority or optimality is claimed in the sense of the new definitions, it would be useful to refer the reader to the definition, as it is not clear whether superiority is meant as a technical term, or simply as a qualitative assessment. (e.g. line 204)

**Limitations:**

See weaknesses.

---

> ### Author Rebuttal · Authors · 2024-08-06
>
> We thank Reviewer TcMZ for the careful reading and the constructive review. We acknowledged that this paper uses conventional notations from the GNN theory literature, such as hash function, without detailed explanations. We will definitely improve the clarity of the writing and the readability of the paper. Below, we carefully address your main concerns.
>
> ---
>
> **Q1.** This is particularly true for superiority proofs, which clearly assume something about the hash function, but nothing is ever stated in regards to it.
>
> **A1.** We note that the hash function in our paper is consistent with the usage in the GNN theory literature, where the hash function is often directly used without further explanation (see, e.g., the seminal work [1] among many others). As a common concept in computer science, in a hash function, identical inputs lead to identical outputs, while different inputs result in distinguishable outputs. In our experiments, we use $\operatorname{hash}(h_i,\\{\\!\\!\\{h_j\\}\\!\\!\\})=h_i+\sum h_j^3$, though it could be substituted with other choices. We will clarify this in our experiments section.
>
> **References:**
>
> [1] Xu, Keyulu, et al. "How powerful are graph neural networks?" ICLR 2019.
>
> ---
>
> **Q2.** In addition to that, it is not immediately clear where the ‘canonization’ view can be useful, beyond the example provided in this paper.
>
> **A2.** We note that canonization is a generic method, and it can be applied to scenarios beyond those demonstrated in our paper. The major advantage of canonization is two folds.
>
> - Theoretically, it enables us to have an essential view of frames and frame averaging, a formal hierarchy of frame complexity, and a principled path to designing optimal frames.  With this theory, we can show the non-universality of SignNet, which is a critical open problems.
> - Empirically, we show that canonization can be applied in many different scenarios with different types of symmetries:
>     - **sign and basis equivariance**: we design optimal or better algorithms (OAP) for sign and basis equviariance of eigenvectors, and show clear benefits in efficiency and performance.
>     - **permutation equivariance**: in the EXP experiment (Section 5.1), we use canonization to achieve permutation equivariance on graph data (not sign/basis invariance of eigenvectors).
>     - **rotation equivariance:** In Appendix D, we applied canonization to achieve rotational equivariance of particles in PCA-frame methods.
>
> ---
>
> **Q3.** First and foremost, while I realize the authors must be approaching from the graph side of the literature - within frame averaging the term used in all papers is ‘canonicalization’, and not ’canonization’, and I would suggest for consistency to refer to it as such, to avoid creating confusion.
>
> **A3.** Thanks for the suggestion. We agree with this and will switch to "canonicalization" in the revision for better consistency.
>
> ---
>
> **Q4.** In subsection 2.1, as soon as frame averaging is presented, an assumption is made about finiteness of the group - but is never stated. Furthermore, after Section 3.2, the vector spaces considered becomes simply finite dimensional Euclidean spaces.
>
> **A4.** This is true. We will clarify these assumptions in our paper.
>
> ---
>
> **Q5.** Line 87 and 95 - universal approximation is claimed, however neither a citation, nor a theorem reference is provided.
>
> **A5.** Thanks for pointing out. We will cite the frame averaging paper after these claims.
>
> ---
>
> **Q6.** Line 128 - why are automorphisms of the input a major obstacle in analyzing complexity of frames?
>
> **A6.** The size of the automorphism group (and thus the frame) depends on the input --- more "symmetric" inputs with larger automorphism group would lead to larger computational complexity in the averaging step. Instead, the size of canonical forms depends less on the input and is more stable. For example, an optimal canonization would have size 1 on all inputs, while the corresponding frame size could be exponential (e.g., for graphs). For the sign ambiguity of LapPE, some eigenvectors become uncanonizable, but their canonization size is still less than or equal to 2, while their corresponding frame sizes could be exponentially large depending on the input.
>
> ---
>
> **Q7.** Line 144 - what you refer to as contractive ‘canonization’, is usually referred to as orbit canonicalization.
>
> **A7**. Thanks for pointing out. We will switch the term to orbit canonicalization for consistency.
>
> ---
>
> **Q8.** In line 205 what is the norm of a vector?
>
> **A8.** On line 205 $|\cdot|$ refers to element-wise absolute value. We apologize for the confusion and will clarify this in the theorem.
>
> ---
>
> **Q9.** In both section 3 and 4 the actual problem being solved is never stated, making it very difficult to follow. By this I mean that given a matrix, you find its eigenvectors, and wish to construct a function which is permutation equivariant …
>
> **A9.** Indeed, we would state the problem more clearly for better readability.
>
> ---
>
> **Q10.** You mention that you have not been able to show that OAP is optimal - given the indices may not exist, how could it be optimal?
>
> **A10.** OAP is optimal iff such indices always exist for **canonizable** inputs. So far we are not able to construct such inputs where these indices do not exist, therefore we are uncertain whether OAP is optimal.
>
> ---
>
> **Q11.** Line 304 - you never say what FA/CA is - I presume this is frame averaging and canonisation averaging?
>
> **A11.** This is true. We defined FA on line 29 and CA on line 146. Nevertheless, we will make sure to clarify this in the main text.
>
> ---
>
> **Q12.** The mentioned typos and suggestions.
>
> **A12.** We will fix them in the revision.
>
> ---
>
> We thank Reviewer TcMZ again for carefully checking our paper and providing many constructive suggestions. We will surely modify our paper according to your suggestions. If you have further concerns, we are happy to address them in the discussion stage.

---

> > ### Comment · Reviewer_TcMZ · 2024-08-09
> >
> > Thanks to the authors for their clarifications and answers. I would like to emphasise that my only concern at this point remains in readability of the paper on two fronts: problem statements and theorem statements. As long as these two are improved in the revision in line with the comments, I would be happy to raise my score.

---

> > > ### Author Response · Authors · 2024-08-10
> > >
> > > We thank Reviewer TcMZ for the prompt response. We clarify our problem statements and theorem statements in the following points, and we will add these discussions to the camera-ready version of our paper for better readability.
> > >
> > > ---
> > >
> > > ### Problem Statements
> > >
> > > Our paper consists of two parts, each addressing different problems. We describe the problems in these sections in detail as follows:
> > >
> > > 1. **Section 3** is focused on the theoretical problem of finding a principled way (i.e., canonicalization) to characterize the complexity of frames $\mathcal{F}(X)$ and frame averaging, a general class of invariant and equivariant learning methods. We have included the definitions of invariant and equivariant learning in Section 2, and the definitions of frames frame averaging in Section 2.1. We will state the main problem more clear in the beginning of Section 3 for better readability.
> > > 2. Guided by the theoretical insights in Section 3, **Section 4** is to design better or optimal canonicalization algorithms for a widely appeared class of problems, the sign and basis invariance of eigenvectors. Specifically, we aim to design a canonicalization algorithm $\mathcal{C}$ operating on eigenvectors $\mathbf{U}\in\mathbb{R}^{n\times d}$, that is **invariant** to sign/basis transformations, **equivariant** to permutation transformations, and outputs a set of eigenvectors $\mathbf{U}^*\in\mathbb{R}^{n\times d}$ in the **same eigenspace** as $\mathbf{U}$. We consider two settings of sign and basis invariance: without (Section 4.1) and with (Section 4.2) permutation equivariance, corresponding to different problem scenarios. We will make it more clear in the beginning too.
> > >
> > > ---
> > >
> > > ### Theorem Statements
> > >
> > > We understand that we define some notations out of the theorems, making them less self-contained and easy-to-understand. To ease your concerns, we will define the notations more clearly, point out the key messages of the main theorem, and add references to previous definitions. Below we give two examples of the modified theorem statements:
> > >
> > > - **Theorem 4.1.** Given a set of eigenvectors $\mathbf{U}\in\mathbb{R}^{n\times d}$, let $\mathscr{P}=\mathbf{UU}^\mathrm{T}$ denote the projection matrix of the eigenspace. Let $\mathbf e_1,\dots,\mathbf e_n$ denote the standard basis vectors. Then, there exists indices $1\leq i_1<\cdots<i_d\leq n$, such that for all $1\leq j\leq d$, we have $\lVert\mathscr P\mathbf e_{i_j}\rVert>0$, and the vectors $\mathscr P\mathbf e_{i_1},\dots,\mathscr P\mathbf e_{i_d}$ are linearly independent.
> > > - **Theorem 4.3.** Let $\alpha_i\ (i=1,\dots,n)$ be the outputs of the hash function in the OAP algorithm defined in Equation (4), and let $i_j\ (j=1,\dots,d)$ be the indices found in Algorithm 3. Then, the MAP algorithm is equivalent to the OAP algorithm by taking $\alpha_i=\lVert\mathscr P_i\rVert$ for all $1\leq i\leq n$ and $i_j=j$ for all $1\leq j\leq d$. The FA-lap algorithm is equivalent to the OAP algorithm by taking $\alpha_i=\mathscr P_{ii}$ for all $1\leq i\leq n$.
> > >
> > > We will modify all the theorems in our paper similarly to enhance readability and self-containedness. These changes will be reflected in the camera-ready revision of our paper. We hope these changes address your concerns. If you have further concerns or suggestions, please feel free to reach out.

---

> > > > ### Author Response · Authors · 2024-08-12
> > > > **Could you please check the revision examples?**
> > > >
> > > > Dear Reviewer TcMZ,
> > > >
> > > > We appreciate your valuable feedback and are committed to revising the paper according to your suggestions. However, as NeurIPS does not allow changes to the PDF during the rebuttal period, we have provided some examples of the revisions we plan to make in the final version. We would greatly appreciate it if you could review these examples to see if they address your concerns satisfactorily.
> > > >
> > > > Thank you for your time and consideration.
> > > >
> > > > Best,
> > > > Authors

---

> > > > > ### Comment · Reviewer_TcMZ · 2024-08-13
> > > > >
> > > > > I would like to thank the author for illustrating the examples of revisions and I would say that these are siginificantly better than their current versions. As long as these are included and overall presentation of the paper is improved, I am happy to raise the score.

---

> > > > > > ### Author Response · Authors · 2024-08-13
> > > > > > **Thanks**
> > > > > >
> > > > > > We will definitely revise the paper according to your suggestions. Thanks again and have a nice day!

---

### Official Review · Reviewer_MgpB · 2024-07-14

**Soundness:** 3
**Presentation:** 4
**Contribution:** 4
**Rating:** 6
**Confidence:** 3

**Summary:**

The work establishes a significant connection between canonicalization and frame averaging, demonstrating an equivalence between the two concepts. By establishing such a relationship, the study efficiently compares the complexity of frames and determines the optimality of frames in relation to the symmetries of eigenvectors. This guides the authors in designing novel frames for eigenvectors that are superior to existing methods, achieving optimality in certain simpler cases. These new frames are both theoretically sound and empirically validated, revealing equivalences between previous methods that had not been identified before. The authors conducted experiments showing the proposed frames' effectiveness on benchmark datasets, achieving higher performance.

**Strengths:**

1. The paper is well-written and nicely explained.

2. The paper provides strong theoretical results.

3. Empirical analysis backs the theoretical reasoning for the proposed architecture achieving higher performance.

**Weaknesses:**

I found the empirical evaluation to be the weak point of the work. The work provides strong theoretical results. However, a more detailed empirical evaluation would make the work more *complete*. For example, considering another molecule dataset (e.g., Alchemy) and texture reconstruction task (section 4.3 SignNet and Basisnet)

1. Line 98: $G_X$ is not define before. I think it should be defined here instead of Line 129
2. proof of theorems in the supplementary is not in order of the main text.

**Questions:**

1. Line 138: “ This converts the problem of finding a G-equivariant subset of the group to finding a G-invariant set of inputs.” - further explanation of this would be great.

2. Line 175: “The canonization size |C(X)| may differ for different inputs.” — what does the size of |C(X)| tell about the input X?

**Limitations:**

The authors discussed the limitations.

---

> ### Author Rebuttal · Authors · 2024-08-06
>
> We thank Reviewer MgpB for the constructive review. We address your concerns as follows.
>
> ---
>
> **Q1.** A more detailed empirical evaluation would make the work more *complete*. For example, considering another molecule dataset (e.g., Alchemy) and texture reconstruction task (section 4.3 SignNet and Basisnet)
>
> **A1.** We provide experiment results on Alchemy in the following table. Due to the limited time of the rebuttal period, we directly followed the SignNet setting and did not tune the hyperparameters for our method, so there is room for improvement. As shown in the table, SignNet, MAP, and OAP all achieve comparable performance. Unfortunately, we are unable to reproduce the texture reconstruction task in SignNet since it was a private code and the authors did not release it (see the [SignNet repo](https://github.com/cptq/SignNet-BasisNet)). Following your suggestion, we will add the full experiment results in our paper to make it more complete.
>
> | Model | Test MAE |
> | --- | --- |
> | GIN | 0.180 ± 0.006 |
> | SignNet | 0.113 ± 0.002 |
> | MAP | 0.114 ± 0.0007 |
> | OAP | 0.115 ± 0.001 |
>
> ---
>
> **Q2.** Line 98: $G_X$ is not define before. I think it should be defined here instead of Line 129. Proof of theorems in the supplementary is not in order of the main text.
>
> **A2.** Thanks for pointing this out. We will define $G_X$ before line 98 and adjust the order of proofs in the appendix.
>
> ---
>
> **Q3.** Line 138: “ This converts the problem of finding a G-equivariant subset of the group to finding a G-invariant set of inputs.” - further explanation of this would be great.
>
> **A3.** Thanks. Here is a more elaborate explanation. In frame averaging, we need to design a "frame" $\mathcal{F}$ that is a subset of the whole group $G$ and is **equivariant** to the group actions in $G$. Theorem 3.1 gives the equivalence of frame and canonization, which allows us to convert the problem of finding a *frame* into finding a *canonization*. In canonization, for each input, we need to find a subset of the input space that is **invariant** to the group actions. This set is called the canonical form of the input. Doing so has several advantages:
>
> - Firstly, since equivariance is a strong requirement, it is often easier to directly construct an invariant canonical form instead of an equivariant frame.
> - Secondly, as proved in Theorem 3.1, the canonization size is $|G_X|$ times smaller than the frame size, making canonization more efficient than frames.
> - Thirdly, canonization theory allows us to characterize the existence of uncanonizable elements when further equivariance constraints are imposed, which further allows us to solve the open problem of the expressivity of SignNet.
>
> We will add this elaboration in the revision.
>
> ---
>
> **Q4.** Line 175: “The canonization size |C(X)| may differ for different inputs.” — what does the size of |C(X)| tell about the input X?
>
> **A4.** The canonization size $|\mathcal{C}(X)|$ reveals **the extent of symmetry (w.r.t. group** $G$) **of the input**, in that **more symmetric inputs often lead to a larger canonization size**. For example, more symmetric graphs with more automorphism (e.g., a regular graph) often result in a larger canonization size. Specifically, the canonization sizes of graphs in the ZINC molecular datasets are mostly within the order of $10^3$, while in the more symmetric EXP dataset, the canonization sizes are in the order of $10^{19}$. We will elaborate on this point in the revision.
>
> ---
>
> We hope our response addresses your concerns. If you have further concerns, we will be happy to address them during the discussion period.

---

> > ### Comment · Reviewer_MgpB · 2024-08-12
> >
> > Thanks for the response.

---

### Official Review · Reviewer_y3Wh · 2024-07-14

**Soundness:** 3
**Presentation:** 3
**Contribution:** 3
**Rating:** 6
**Confidence:** 3

**Summary:**

This work makes connections between two model-agnostic approaches to designing equivariant networks: frame averaging and canonization. It is first shown that any function obtained using frame-averaging can also be obtained using canonization. Then it is shown that canonization is computationally more efficient than frame averaging. Further, it is shown that not all elements are "canonizable", i.e., some elements may not have a unique canonical form. This insight helps prove that SignNet and BasisNet are not universal, and they further propose Orthogonalizes Axis Projection (OAP) that leads to optimal designs for the problem of sign and basis equivariance when permutation equivariance is not required. The optimality of the case when permutation equivariance is required remains unsolved. Experiments on several graph datasets confirm validate the working of the proposed OAP method.

**Strengths:**

- The paper is well-written.
- The connection between canonization and frame averaging is very interesting. It helps resolve the issue of universality of sign and basis networks. Further, for the case without permutation, a universal algorithm is also proposed.
- Experimental results show that the proposed method is expressive and computationally more efficient than frame averaging.
- Experimental results on various graph datasets show superior performance to prior methods.

**Weaknesses:**

- Could you please provide a comparison of compute memory and time for all the networks in the experiments. Especially, comparison with non-frame-averaging methods such as GIN would give better insights to the readers on their applicability. Currently, it is not clear how much more expensive/cheap FA/canonicalization is compared to baselines such as GIN.
- In Tab. 2, the practical advantage of canonization over frame averaging in computational complexity for graph dataset seems negligible because of the huge absolute compute time. In Tab 2., how is the averaging done over such a large set? Please provide any preprocessing time required for such computations as well (if applicable).
- Although the initial results in the paper are applicable to the general area of equivariant learning, the applications of the initial results seems to be only useful in the context of equivariance to sign and basis. But the title of the paper has no mention about sign and basis, which makes it look more generally useful than what the experiments indicate.

**Questions:**

- In Tab. 1, out of curiosity what happens if we use only GIN+ID? I am curious because GIN+ID is universal (unlike GIN), so, the results on GIN+ID would help understand the benefit of permutation equivariance. Also, some results/plots on the convergence benefits from equivariance would help.
- Are there any other domains where the results on the connection between frame averaging and canonization is directly applicable?

**Limitations:**

Yes.

---

> ### Author Rebuttal · Authors · 2024-08-06
>
> We thank Reviewer y3Wh for the constructive review. We address your concerns as follows.
>
> ---
>
> **Q1.** Could you please provide a comparison of compute memory and time for all the networks in the experiments. Especially, comparison with non-frame-averaging methods such as GIN would give better insights to the readers on their applicability. Currently, it is not clear how much more expensive/cheap FA/canonicalization is compared to baselines such as GIN.
>
> **A1.** Yes! Methodologically, the proposed canonization algorithm is a pure preprocessing algorithm that does not increase the training time, and the pre-processing time is negligible compared with training time. On the other hand, FA methods such as SignNet increase the training time. The canonization algorithm also has no influence on the memory.
>
> We compare the time and memory of canonization methods with their non-FA backbone in the following table. Using canonization algorithms only increases the pre-processing time of the backbone, which is negligible compared to the training time. On the other hand, the two-branch architecture of SignNet increases the training time and memory. We will include all compute time and memory statistics in our paper.
>
> | Model | Pre-processing time | Training time | Total Time | Memory |
> | --- | --- | --- | --- | --- |
> | GatedGCN backbone | - | 3h26min | 3h26min | 1860MiB |
> | GatedGCN + SignNet | 30.03s | 4h13min | 4h13min | 2124MiB |
> | GatedGCN + MAP | 133.67s | 3h20min | 3h22min | 1850MiB |
> | GatedGCN + OAP | 186.38s | 3h25min | 3h28min | 1860MiB |
> | PNA backbone | - | 16h31min | 16h31min | 2242MiB |
> | PNA + SignNet | 30.03s | 18h1min | 18h1min | 2570MiB |
> | PNA + MAP | 133.67s | 16h47min | 16h49min | 2244MiB |
> | PNA + OAP | 186.38s | 14h54min | 14h57min | 2312MiB |
>
> ---
>
> **Q2.** In Tab. 2, the practical advantage of canonization over frame averaging in computational complexity for graph dataset seems negligible because of the huge absolute compute time. In Tab 2., how is the averaging done over such a large set? Please provide any preprocessing time required for such computations as well.
>
> **A2.** Since the frame and canonization sizes are both extremely large, we need to subsample them in practice. For fair comparison, we adopt the same subsampling size. In this case, **the advantage of canonization at the averaging complexity translates into a better sample efficiency for approximating the averaging expectation, which leades to faster training convergence, as we proved in Appendix C**. As for computing the frame/canonization of the same size, frame/canonization takes 2.53s and 3.15s for pre-processing, whose difference is negligible compared to the training time of 67.37s.
>
> ---
>
> **Q3.** Although the initial results in the paper are applicable to the general area of equivariant learning, the applications of the initial results seems to be only useful in the context of equivariance to sign and basis. But the title of the paper has no mention about sign and basis, which makes it look more generally useful than what the experiments indicate.
>
> **A3.** We note that our canonization perspective established in Section 3 is generic and applicable to different types of equivariance. In Section 4, we focus on sign/basis invariance because they are one of the most challenging problems with exponentially large group sizes. Apart from that, we also applied our canonization algorithms to other symmetries, including:
>
> - **permutation equivariance** of graph data in the EXP experiment;
> - **rotational equivariance** of particles In Appendix D.
>
> These new applications illustrate the generality of our analysis and the proposed algorithms, which leads to the use of a more general title. We will elaborate more on this part to avoid any confusion. Thanks!
>
> ---
>
> **Q4.** In Tab. 1, out of curiosity what happens if we use only GIN+ID? I am curious because GIN+ID is universal (unlike GIN), so, the results on GIN+ID would help understand the benefit of permutation equivariance. Also, some results/plots on the convergence benefits from equivariance would help.
>
> **A4.** Following your suggestion, we evaluate GIN with two kinds of commonly used node IDs: one-hot node ID and random node features.
>
> 1. Models with only one-hot node IDs is **not robust**: when we apply a random permutation to the input graph during training, the model fails with only 0.5 accuracy, while FA-GIN+ID is not affected. This shows that without permutation equivariance, the model is not able to learn real structural information.
> 2. Models with only random features **converges much slower** than the deterministic FA-GIN+ID model (see a training progress comparison in Figure 3 of [1]). This is because it’s hard to learn from random features without exact permutation symmetry.
>
> The above results show that lacking permutation equivariance hurts the robustness and convergence speed of GNNs. Real-world tasks are much more complex than EXP, thus lacking equivariance could harm the performance of GNN models significantly.
>
> **References:**
>
> [1] Ma et al. Laplacian canonization: A minimalist approach to sign and basis invariant spectral embedding. *NeurIPS 2023*.
>
> ---
>
> **Q5.** Are there any other domains where the results on the connection between frame averaging and canonization are directly applicable?
>
> **A5.** Since our analysis applies to general groups, the connection applies to any type of equivariance. In this paper, we have explored **sign/basis invariance, permutation equivariance, and orthogonal equivariance**. In future work, it is also possible to apply frame and canonization to the rotational equivariance of point clouds/molecules, the permutation equivariance of multisets, etc. Therefore, we believe that canonization would serve as a new generic perspective for understanding and attaining equivariance.
>
> ---
>
> We hope our response addresses your concerns. If you have further concerns, we will be happy to address them during the discussion period.

---

> > ### Comment · Reviewer_y3Wh · 2024-08-11
> > **Thank you for your response**
> >
> > I thank the authors for their clarifications. For readability, please provide references in the main text to the results in the appendix. Overall, I am happy to increase my score since most of my concerns are clarified. I still find the applications/usefulness limited and the title of the paper seems more general than the applications.

---

> > > ### Author Response · Authors · 2024-08-12
> > > **Thanks**
> > >
> > > We are glad to hear that our responses have addressed your concerns. We will incorporate references in the main text to the results in the appendix and revise the title to more accurately reflect the content of the paper in the revision. Thank you again and have a good day!

---

### Decision · Program_Chairs · 2024-09-25

**Decision:**

Accept (poster)

**Comment:**

After going through the reviews and author’s rebuttal, I find that the reviewers unanimously agree that this paper presents a valuable contribution by establishing a connection between frame averaging and canonization. This connection leads to a better understanding of frame design and the development of improved frames, particularly for sign and basis equivariance. The theoretical results are strong and well-supported by empirical evidence. The reviews also generally agree that the paper is well written.

However there were some concerns on presentation of the problem statements and theorem statements which seems to have been addressed in the rebuttal responses. The empirical evaluation also seems to be an area of weakness, specially over the limited score of application. Additional dataset results and discussions over multiple application scenarios of canonization seems to address this to a certain extent.

Overall I would **recommend acceptance of this paper with minor revisions**. The paper presents a valuable contribution to the field of equivariant and invariant learning. The theoretical results are strong, and the empirical evidence supports the effectiveness of the proposed methods.

Addressing the minor concerns, such as adding discussion on scope of application and including empirical evaluations from rebuttal, will further strengthen the paper's impact and make it a valuable addition to the conference. Here are some specifics to be taken care of -
1. Please include time and memory statistics as discussed in response to reviewer y3Wh
2. Empirical Evaluation: Please Include additional datasets results as discussed in response to reviewr MgpB.
3. Presentation concerns: As discussed in response to reviewer TcMZ, please make necessary changes in problem and theorem statements.
4. Please include discussions with reviewer 8tUB over application scope in the final version.

Also, please consider revising the title to more accurately reflect the paper's specific focus, as well as address the remaining reviewer concerns, such as clarifying certain theoretical points, minor typos and including clarifying discussions.